Evidence synthesis  

ecology, behaviour, ecosystems

bioenergetics, anthropogenic disturbance, life-history traits, population dynamics, risk assessment, marine mammals

**Author for correspondence:**
Kelly A. Keen
e-mail: kelly.a.keen@gmail.com

# Emerging themes in Population Consequences of Disturbance models

Kelly A. Keen[1], Roxanne S. Beltran[1], Enrico Pirotta[3,4] and Daniel P. Costa[1,2]

[1]Department of Ecology and Evolutionary Biology, and [2]Institute of Marine Sciences, University of California, Santa Cruz, CA, USA
[3]Centre for Research into Ecological and Environmental Modelling, University of St Andrews, UK
[4]School of Biological, Earth, and Environmental Sciences, University College Cork, Cork, Ireland

(iD) KAK, 0000-0001-7839-6875; RSB, 0000-0002-8520-1105; EP, 0000-0003-3541-3676; DPC, 0000-0002-0233-5782

Assessing the non-lethal effects of disturbance from human activities is necessary for wildlife conservation and management. However, linking short-term responses to long-term impacts on individuals and populations is a significant hurdle for evaluating the risks of a proposed activity. The Population Consequences of Disturbance (PCoD) framework conceptually describes how disturbance can lead to changes in population dynamics, and its real-world application has led to a suite of quantitative models that can inform risk assessments. Here, we review PCoD models that forecast the possible consequences of a range of disturbance scenarios for marine mammals. In so doing, we identify common themes and highlight general principles to consider when assessing risk. We find that, when considered holistically, these models provide valuable insights into which contextual factors influence a population's degree of exposure and sensitivity to disturbance. We also discuss model assumptions and limitations, identify data gaps and suggest future research directions to enable PCoD models to better inform risk assessments and conservation and management decisions. The general principles explored can help wildlife managers and practitioners identify and prioritize the populations most vulnerable to disturbance and guide industry in planning activities that avoid or mitigate population-level effects.

## 1. Introduction

A significant hurdle for wildlife conservation and management is knowing when and how short-term responses to human activities result in biologically meaningful changes that affect population dynamics [1]. For many vertebrates, human activities may elicit behavioural (e.g. avoidance, reduced foraging and changes in vocalizations) and physiological (e.g. increased stress levels and temporary reductions in hearing) responses that can, in the aggregate, affect individual fitness and cause population- and ecosystem-level effects [2–4]. Assessing the consequences of anthropogenic disturbance is therefore necessary for the long-term persistence of populations in increasingly human-altered ecosystems [5]. For marine mammals, this is a requirement for most risk (or impact) assessments under European Union (Habitats Directive 92/43/EEC) and United States (Marine Mammal Protection Act (MMPA), 16 U.S.C. §§ 1361 *et seq*.) legislation [6]. However, the links between behavioural and physiological responses and their long-term individual- and population-level effects are poorly understood, making comprehensive risk assessments difficult [2,7].

To address this issue, the Population Consequences of Disturbance (PCoD) framework was developed to conceptualize how disturbance-induced changes in individual behaviour and physiology affect population dynamics via changes in individual health and vital rates (electronic supplementary material, figure S1) [7,8]. While this framework was developed for use with marine mammals, it is generally applicable across most vertebrates. The PCoD framework separates

the behavioural and physiological effects of disturbance into acute and chronic impacts. Acute impacts directly affect an individual's vital rates by, for example, increasing their risk of predation or decreasing their probability of survival following injury (e.g. collision with a vessel and entanglement in fishing gear). By contrast, chronic impacts result from persistent or recurring activities that affect an individual's health over a prolonged period (e.g. seismic surveys and whale watching), potentially reducing lifetime reproductive output. An individual's health is related to many aspects of its physiology, including stress levels, immune status and energy stores. While our understanding of marine mammal stress physiology and immunology is incomplete [9], bioenergetics has proven to be a useful approach for implementing the PCoD framework [7,10]. Consequently, most implementations to date have focused on changes in a female's time-energy budget concerning lost foraging time, the subsequent effects on energy delivery from mother to offspring and the cascading long-term impacts on the population [11–13].

Since the PCoD framework was first proposed, a suite of models has been created to evaluate the short- and long-term consequences of disturbance [2]. These models take a quantitative approach to the PCoD framework using a combination of matrix modelling [14], physiologically structured population modelling [15], bioenergetic modelling [10] and stochastic dynamic programming [16]. PCoD models have been developed for several marine mammal species and parametrized via species-specific empirical data and alternative methods, including extrapolating from other species [17], proxy relations [18], expert elicitation [19] and informed assumptions [20], when empirical data are lacking (see [2] for how empirical data and alternative methods have been used to parametrize PCoD models). PCoD models have been used to forecast the possible consequences of a range of disturbance scenarios and identify the disturbance level likely to result in a population impact [21]. As such, PCoD model outputs can provide valuable insights into what contextual factors influence a population's likelihood and duration of exposure and sensitivity to disturbance. However, the findings from these disparate models have yet to be synthesized in a single review to guide risk assessments for marine mammals. This information can help wildlife managers and practitioners identify and prioritize the populations most vulnerable to disturbance (see decision framework in [22]) and guide industry in planning activities that avoid or mitigate population-level effects.

In this synthesis, we review common themes that have emerged from these disparate PCoD models. In so doing, we highlight essential intrinsic and extrinsic factors to consider when assessing risk and describe how they can be evaluated when determining a population's degree of exposure and sensitivity to disturbance. We also identify data gaps and suggest future research directions that will enable PCoD models to better inform risk assessments and conservation and management decisions. Finally, in the electronic supplementary material, we discuss how the PCoD framework and the emerging themes in this synthesis can be broadly applied to guide risk assessments for other species (see *Applicability to other species*), as well as underlying model assumptions and limitations that can influence model predictions but may not be self-evident to non-specialists or non-statisticians (see *PCoD model assumptions and limitations*).

We searched for PCoD model publications from 2005 (when the PCoD framework was first proposed [8]) through March 2021 on Google Scholar using the search terms 'marine mammal', 'bioenergetics', 'model', 'population consequences' and 'disturbance'. We located additional publications by searching the references cited in a review paper by Pirotta *et al.* [2] and the citing literature of each publication that met our synthesis criteria. To meet our criteria, the publication had to quantify the non-lethal effects of anthropogenic disturbance on marine mammal vital rates via the behavioural–bioenergetic pathway. The publications included in the synthesis are provided in the electronic supplementary material, table S1.

## 2. Life-history traits

When assessing the risk associated with a proposed activity, it is essential to determine (i) if the population will be exposed, (ii) the proportion of the population exposed, (iii) the duration of individual exposure and (iv) the sensitivity of the exposed individuals [22,23]. The answers to these questions are influenced by the life-history traits of an individual or population, including their movement ecology, reproductive strategy, body size and pace of life. In the following sections, we examine the importance of each life-history trait for mediating risk (figure 1). A summary of the life-history traits exhibited by marine mammals (at the family level) is provided in the electronic supplementary material, tables S2 and S3.

### (a) Movement ecology

A population's movement ecology influences its degree of exposure to a disturbance-inducing activity [18,24–26]. Marine mammal movement patterns broadly fall into three categories: resident, nomadic and migratory (figure 2*a*; electronic supplementary material, table S2). Individuals that exhibit resident movement patterns (e.g. sea otters (*Enhydra lutris*) [27]) occupy small home ranges relative to the population's overall range [28], whereas nomadic individuals (e.g. spinner dolphins (*Stenella longirostris*) [29]) move over much of the population's range without spatial or temporal consistency [30]. Consequently, the same disturbed area could frequently expose a few individuals of a resident population while infrequently exposing many individuals of a nomadic population (figure 2*a*) [25,31]. However, the proportion of the population exposed will depend on the spatial extent of the disturbance relative to the population's range. Because foraging grounds and reproductive areas spatially overlap for both resident and nomadic populations, PCoD modelling shows that the behaviours potentially disrupted will depend on the timing of the disturbance event [32].

By contrast, individuals that exhibit migratory movement patterns transit annually or seasonally between sites within their range [30,33], which reduces the potential for year-round, prolonged exposure, as simulated in several PCoD models (figure 2*a*) [13,18,25]. Some migratory populations (e.g. southern elephant seals (*Mirounga leonina*) [34]) have spatially (e.g. by thousands of kilometres) and temporally (e.g. by several months) separate foraging grounds and reproductive areas. Consequently, if a disturbance-inducing activity occurs in reproductive areas when the population is at the foraging grounds, the likelihood of exposure is zero unless it has lasting environmental effects (e.g. Deepwater Horizon oil spill in the Gulf of Mexico [35]). The reverse scenario is also generally true, but some individuals may remain at the foraging grounds during the reproductive season [36]. Other

**Figure 1.** Emerging themes in PCoD models that should be considered when assessing the likelihood and duration of exposure and the sensitivity of a population to disturbance. (Online version in colour.)

migratory populations do not have separate foraging grounds and reproductive areas and instead migrate in response to seasonal ecological conditions, such as advancing sea ice and migration of prey (e.g. beluga whales (*Delphinapterus leucas*) [37]). For these populations, PCoD modelling shows that foraging and/or reproductive behaviours may be disrupted if there is spatial and temporal overlap [13].

Migratory populations may also be exposed to anthropogenic disturbance within migratory corridors. While the proportion of the population exposed will be high during synchronous migrations, individual exposure levels will be relatively low because individuals generally transit quickly [26]. Recent studies on the behavioural responses of migrating humpback whales (*Megaptera novaeangliae*) to seismic airgun noise found that individuals deviated from their predicted path and slowed their progression until the source had

passed or ceased [38,39]. Because humpback whales rely on finite energy stores during migration, these behavioural changes could alter a female's energy budget, thereby reducing calf growth rates, and delay arrival at the foraging grounds [40]. When incorporated into a PCoD model, Dunlop *et al.* [41] found that similar behavioural responses to a simulated 10-day seismic survey during peak migration had negligible effects on female body condition and population growth. However, the costs of repeated exposures may accumulate over the long migration, particularly for populations migrating along coastlines with high levels of human activity.

Although not explicitly included in PCoD models to date, it is also necessary to consider whether migratory populations share common foraging grounds or reproductive areas when assessing population-level effects. This will influence whether a single population, a fraction of a population

Proc. R. Soc. B 288: 20210325

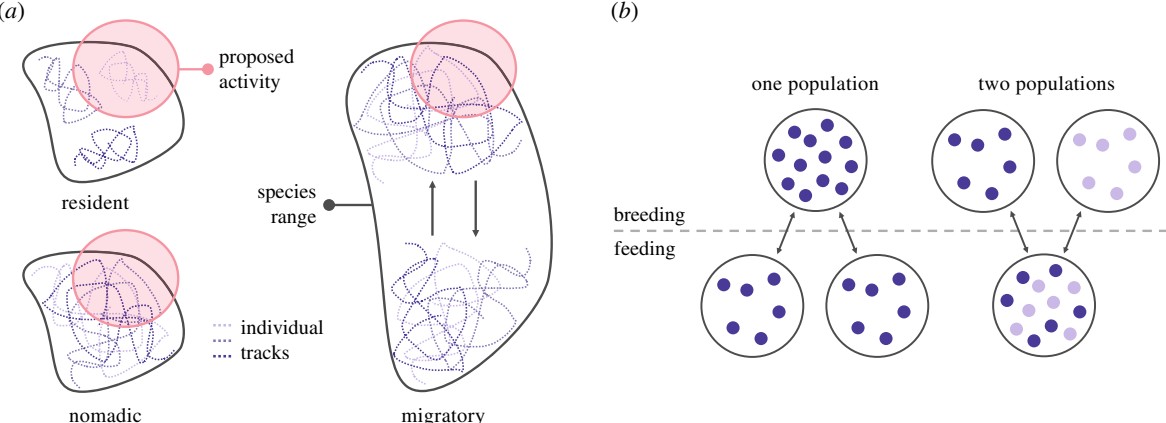

**Figure 2.** (*a*) A population's movement patterns can help determine the degree of exposure to a disturbance-inducing activity. When comparing resident and nomadic movement patterns, the same disturbed area could frequently expose a few resident individuals while infrequently exposing many nomadic individuals. By contrast, many individuals of a migratory population could be exposed, but exposure would be seasonal. Figure adapted from Costa *et al.* [25]. (*b*) Migratory populations may share common foraging grounds or reproductive areas, which influence whether a single population, a fraction of a population or multiple distinct populations will be exposed to a disturbance-inducing activity. If individuals share a common reproductive area but return to unique foraging grounds, individuals constitute one population representing one demographic breeding unit. Alternatively, if they share a common foraging ground but return to unique reproductive areas, they represent two or more demographically distinct populations. (Online version in colour.)

or multiple distinct populations are exposed (figure 2*b*). If individuals share a common foraging ground but return to unique reproductive areas, they represent two or more demographically distinct populations. Alternatively, if they share a common reproductive area but return to unique foraging grounds, individuals still represent one demographic breeding unit [42]. Consequently, a disturbance-inducing activity within a foraging ground used by more than one breeding unit will affect more than one population. Distinct breeding units may also overlap spatially but not temporally [43]. Thus, disturbance-inducing activities that spatially overlap with these reproductive areas may impact one or more populations depending on the activity's temporal extent.

Individual movement patterns are also influenced by demographic factors, including the age, sex and reproductive status of individuals within a population. For example, demographic differences may impact where individuals forage [44], when individuals migrate [45] and how individuals aggregate [46]. Identifying which population segments will be affected can benefit mitigation and activity planning by reducing or avoiding interactions with sensitive individuals (see *Reproductive strategy* and *Body size*).

Understanding a population's movement ecology can help determine the likelihood and duration of exposure and the proportion of the population exposed. Biologging devices and visual and acoustic survey methods provide valuable information on marine mammal movements [47] that can be incorporated into risk assessments to help identify which populations may be present in a disturbed area and thus require further assessment [22]. This information can also be used to plan activities that avoid areas with high concentrations of marine mammals. For example, Hückstädt *et al.* [48] used tracking data for several humpback whale and blue whale (*Balaenoptera musculus*) populations to identify which were particularly susceptible to exposure from seismic surveys and where surveys could have the largest impacts. PCoD models have been developed to be spatially explicit, using both coarse- [13] and fine-scale [49] movement data, or spatially implicit, with movement data reflected in activity budgets [12] or not included at all [15]. Ultimately, the scale of movement

necessary to assess risk depends on the target population and the management or policy issue being addressed.

## (b) Reproductive strategy

Reproduction is the most energetically expensive period of a female's life cycle because she must balance her own needs with those of her dependent offspring. Strategies for financing reproductive costs are often described on a continuum from income breeding, where energy is acquired throughout lactation, to capital breeding, where energy for lactation is stored as endogenous reserves prior to parturition [50,51]. A population's position on this continuum can influence its sensitivity to disturbance (electronic supplementary material, table S2). For example, the need for income breeders to feed during lactation strongly ties reproductive success to local prey abundance [52], leaving females particularly vulnerable to prolonged foraging loss due to disturbance during lactation [53]. As such, these sensitivities to disturbance in income breeders can lead to declines in offspring recruitment and overall population size [12]. By contrast, capital breeders are less sensitive to short-term foraging losses, particularly during the lactation period, because they rely on energy that has already been stored [13,52]. This buffer may also provide capital breeders with the ability to adjust their foraging behaviour and seek out productive prey patches or alternative prey sources [52]. However, energy stores are finite, and if disturbance is prolonged, particularly within important foraging areas (see *Disturbance source characteristics*) [13], or if the available foraging habitat is spatially limited [25], capital breeders may not be able to accumulate sufficient energy for successful reproduction.

PCoD models provide a framework for quantifying the baseline energetic requirements for reproduction and survival and simulating the downstream impacts of compromised foraging success on population growth. These models show that reduced foraging opportunities can delay sexual maturity or age at first reproduction [15,54,55] and increase the interval between reproductive events [15,54], which could impact a female's lifetime reproductive output and, ultimately,

Proc. R. Soc. B **288**: 20210325

population abundance. Reduced foraging may also decrease the energy transferred to offspring, resulting in reduced fetal growth [56] and/or lower weaning mass, which could affect offspring survival [11–13]. If foraging or energy reserves are reduced such that females struggle to meet their own metabolic demands, the termination of gestation or lactation may also occur [12,13,15,57]. Ultimately, a series of physiological thresholds are reached when a female lacks sufficient resources to reproduce or transfer energy to offspring, which may lead to the termination of an existing reproductive attempt [58,59]. However, how and when these thresholds are reached is poorly understood.

PCoD models identify lactation as the most sensitive reproductive state for income breeders because energy is acquired during lactation to support dependent offspring [12,17]. By contrast, PCoD models identify pregnancy as the most sensitive reproductive state for capital breeders because energy reserves are accumulated during pregnancy to support lactation [11,17]. Model simulations have also demonstrated that the timing of a disturbance-inducing activity during these sensitive states influences whether a female can compensate for reduced or lost foraging. For income-breeding California sea lions (*Zalophus californianus*), simulations carried out by McHuron *et al.* [60] found that the costs associated with nursing a pup were much greater during late lactation than early lactation because the total energy delivered to the pup increased as the pup grew. By contrast, for capital-breeding northern elephant seals (*Mirounga angustirostris*), Pirotta *et al.* [61] found that simulated females could better compensate for disturbance during the first phase of their 8-month foraging trip if the disturbance was not severe. However, irrespective of reproductive strategy, extrinsic factors related to the disturbance source (see *Disturbance source characteristics*) and environment (see *Environmental conditions*) will also influence the outcome of an existing reproductive attempt.

The reproductive strategies exhibited by marine mammals can help assess a population's sensitivity to disturbance and should be considered in risk assessments. Reproductive cycle plots can help identify the temporal overlap between sensitive life-history events and a proposed activity (figure 3) [22]. Management plans that limit or avoid interactions with sensitive reproductive states are ideal. However, as this is not always possible, limiting interactions by avoiding biologically important habitats or reducing the duration and frequency of disturbance-inducing activities may provide individuals with an opportunity to avoid or compensate for the disturbance (see *Disturbance source characteristics*). Additionally, when preparing a risk assessment for a long-term activity, environmental conditions that affect prey availability and marine mammal distribution should be considered when known (see *Environmental conditions*).

## (c) Body size

Body size profoundly influences marine mammal life-history strategies because it affects the rate at which energy is acquired from the environment and how it is allocated to growth, reproduction and survival [62]. While an individual's absolute metabolic rate increases with increasing body size, its mass-specific metabolic rate decreases [63]. This negative allometry between body size and metabolic rate offers a suite of benefits for large marine mammals [64], including a lower cost of transport [65] and enhanced fasting [66] and diving

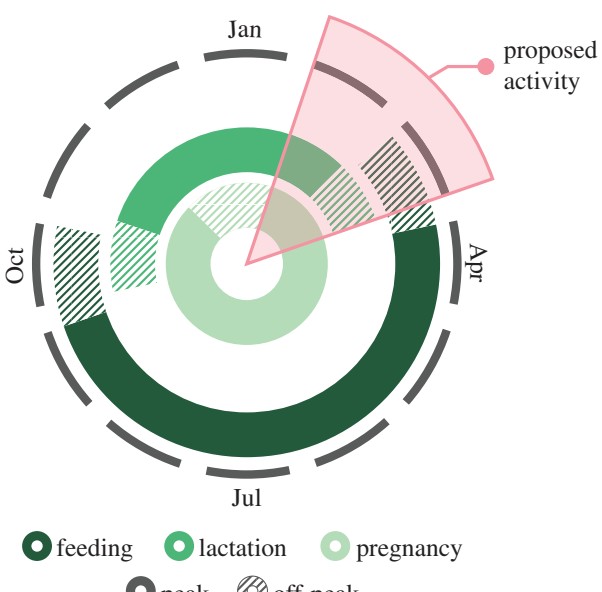

**Figure 3.** A reproductive cycle plot for a North Atlantic minke whale (*Balaenoptera acutorostrata*) showing the variability in the timing and duration of life stages for a 1-year reproductive cycle. The wedge represents a proposed disturbance-inducing activity occurring for approximately 2 months. Reproductive cycle plots can help identify the temporal overlap between sensitive life-history events and a proposed activity. Figure adapted from Wilson *et al.* [22] and informed based on Christiansen *et al.* [56]. (Online version in colour.)

abilities [67]. Smaller individuals or species expend more energy per unit mass than larger ones and thus require a relatively higher resource acquisition rate to meet their metabolic demands. These species, such as sea otters and harbour porpoises (*Phocoena phocoena*), are generally non-migratory or exhibit an income-breeding strategy, relying on concentrated, predictable prey year-round. By contrast, larger species, such as most mysticetes (i.e. baleen whales; suborder Mysticeti), have lower mass-specific metabolic rates that result in a more economical lifestyle. These species are generally migratory and exhibit a capital-breeding strategy, as large body sizes enable them to undergo long migrations while fasting due to their increased energy storage capacity.

Within the context of disturbance, larger body size may be valuable in buffering against periods of reduced food availability [66,68]. For example, PCoD modelling showed that blue whales may be able to compensate for periods of decreased food availability due to their greater capacity to store energy, which allows them to fast for extended periods [13,69]. By contrast, the small size of harbour porpoises may require them to forage nearly continuously to meet their energy needs [70] (but see [71]). As a result, even a moderate disturbance that disrupts foraging or increases energy expenditure could have severe fitness consequences [72]. However, when prey is reduced or limited, smaller-bodied species may be better able to meet their energetic needs than larger ones because they require less food in total [64]. Interactions between body size, reproductive strategy and movement ecology ultimately add complexity to these general patterns. For example, sperm whales (*Physeter macrocephalus*) may be less resilient than whales of similar size due to their income-breeding strategy coupled with the physiological properties of their blubber (e.g. the vast majority of their blubber lipids are stored

as wax esters, a less accessible source of metabolizable energy, rather than as triacylglycerols [73]). As a result, PCoD model outputs suggest that sperm whales may be poorly adapted to handle foraging disruptions [55].

The physiological constraints of body size may also affect how individuals in early life stages respond to disturbance. Among all life stages, PCoD models often identify juveniles and younger (i.e. smaller) mature females as the most sensitive groups within a population [13,15,55,74]. Their small body size limits the amount of energy that can be stored and, subsequently, their ability to compensate for reduced or lost foraging opportunities [75,76]. This may be particularly relevant during years of unfavourable environmental conditions or intense disturbance as younger individuals are less experienced and thus less able to respond to environmental stressors. In addition, young marine mammals have physiological limitations such as limited oxygen-carrying capacities that constrain their diving abilities (i.e. dive depth and duration) and limit their behavioural flexibility [77]. Due to their small body size, PCoD models predict that juveniles will be more prone to starvation and younger mature females will be less likely to bring a pregnancy to term or successfully wean offspring [13,15,55,74]. Reduced energy acquisition during this important developmental period can also affect the amount of energy allocated to growth. While individuals may be able to compensate for slowed growth over time, their lifetime reproductive output could be impacted [78].

Understanding which species and life stages may be exposed can help assess which populations may be most sensitive to a disturbance-inducing activity. Among long-lived mammals, juvenile survival and fertility are the vital rates most sensitive to changes in prey availability [79,80]. Thus, a disturbance source that affects foraging may increase juvenile and offspring mortality and impact population dynamics. If the disturbance is severe, adult mortality, which has the strongest influence on population dynamics in long-lived species [79,80], may also be impacted. Therefore, a disturbance event causing changes in adult survival, although more unlikely, has the potential to cause larger population-level effects.

## (d) Pace of life

PCoD models are parametrized with a population's (or related species') life-history traits, such as age at sexual maturity, interbirth interval and lifespan, which determine the intrinsic rate of population growth, a fundamental component of population dynamics. These traits covary on a fast–slow continuum of strategies that describe how individuals allocate resources to growth, reproduction and survival, which sets the pace of life. Thus, populations that exhibit early maturity, high reproductive rates and short lifespans are said to lead a fast pace of life, whereas those characterized by late maturity, low reproductive rates and long lifespans lead a slow pace of life [59,81]. A population with a fast pace of life will thus have a higher growth rate than a population with a slow pace of life, which has implications for how quickly a population can recover after disturbance disrupts vital rates.

There is considerable variation in life-history strategies across marine mammal species and populations (electronic supplementary material, table S3) [82]. Within cetaceans, for example, harbour porpoises reach sexual maturity around 3.5 years, can reproduce annually and generally live for 10 to 15 years [83,84], whereas bowhead whales (*Balaena mysticetus*) become sexually mature around 25 years, reproduce every 3 to 5 years and can live for over 100 years [85]. The pace-of-life traits of otariids (i.e. sea lions and fur seals; family Otariidae) and phocids (i.e. true seals; family Phocidae) are comparatively less varied, with the average age at sexual maturity ranging from 3 to 7 years, a reproductive cycle of 1 year and lifespans typically ranging from 20 to 40 years [86]. For mysticetes and some odontocetes (i.e. toothed whales; suborder Odontoceti), multi-year reproductive cycles mean that a subset of the population is pregnant in any given year. By contrast, most otariids and phocids have an annual reproductive cycle (although there are some exceptions, including Australian sea lions (*Neophoca cinerea*) [87]) but have been shown to breed intermittently, so the possibility of producing one offspring per year is not always realized [88].

On the fast–slow continuum of odontocete life-history strategies, harbour porpoises are an iconic example of 'life in the fast lane' [84]. Due to their high metabolic requirements, reproductive success is directly tied to prey availability, meaning that harbour porpoises will reproduce annually when prey is abundant (or energy-rich) and less often when prey is scarce or absent (or energy-poor) [84,89]. Therefore, any given reproductive event is likely to be more sensitive to reduced or lost foraging. If the disturbance is severe, adult survival may also be impacted [70]. However, due to their high reproductive rate and short generation time, PCoD model simulations predicted that populations should be quick to recover following a disturbance event [49]. By contrast, long-finned pilot whales (*Globicephala melas*) are an example of 'life in the slow lane', with females reaching sexual maturity around 8 to 9 years, after which they reproduce every 5 years, and live for approximately 60 years [90]. Like harbour porpoises, long-finned pilot whales feed year-round and their energy reserves respond rapidly to prey availability [15], but their larger body size provides a buffer between prey patches. Therefore, under a similar disturbance scenario, any given reproductive event will likely be more resilient to reduced or lost foraging for long-finned pilot whales. However, PCoD model simulations predicted that a female's lifetime reproductive output will decrease if the disturbance is severe [15,91]. For these populations, recovery will take longer due to their lower reproductive rate and longer generation time.

Understanding how disturbance may affect vital rates such as fecundity and survival can provide valuable insights into a population's response to a disturbance event [74,92] and guide mitigation and management strategies that target important life-history stages (e.g. mating and reproduction) or specific age classes (e.g. juveniles and adults) [93]. Knowing a population' life-history strategy can also be helpful when comparing the sensitivities of multiple species within a disturbed area [22]. When sufficient data are not available to determine a species' life-history strategy, species with similar paces of life may be appropriate substitutes.

## 3. Disturbance source characteristics

An individual's sensitivity to a disturbance-inducing activity is also affected by the characteristics of the disturbance source. For example, the spatial and temporal features and nature of the disturbance source (e.g. type (sonar), operational characteristics (intensity, frequency), behaviour

(moving, stationary)) can interact with life-history traits and other contextual factors to influence the probability and severity of individual responses [94]. In the following sections, we examine the importance of these disturbance source characteristics when assessing risk and mitigating effects (figure 1).

## (a) Overlap with biologically important habitats

A population's sensitivity to disturbance will be strongly influenced by the importance of the disturbed area for foraging, reproduction and migration, as shown in several PCoD models [13,49,57,61,95,96]. For example, simulations carried out by Pirotta *et al.* [61] found that disturbance within important foraging areas had a more dramatic effect on adult female northern elephant seal energy budgets than a similar disturbance located in less important habitat within the population's range. Thus, by leaving optimal foraging habitat undisturbed, individuals may be able to meet their energy needs even if other portions of their range are disturbed. In simulations with sperm whales, Farmer *et al.* [57] found that even partial closures of important habitat to seismic surveys could nearly eliminate the risk of individuals reaching terminal starvation. Ultimately, the magnitude of any effect will depend on whether similar habitat of sufficient area is available within the population's range, as well as the temporal characteristics (see *Duration and frequency*) and nature of the disturbance source and the exposure context (see *Nature and context*).

The International Union for Conservation of Nature's (IUCN) Marine Mammal Protected Areas Task Force began identifying Important Marine Mammal Areas (IMMAs) to inform wildlife managers of the spatial and temporal extent of habitat necessary for the viability of marine mammal populations [97]. Since IMMAs have yet to be identified for all populations, similar tools (e.g. the Convention on Biological Diversity's Ecologically or Biologically Significant Areas, the United States' and Australia's Biologically Important Areas and the IUCN's Key Biodiversity Areas) can be used in the interim to assess whether important marine mammal habitats overlap with proposed disturbance-inducing activities. For activities that span decades (e.g. offshore wind farms), these static spatial management tools may be less effective unless they consider ecological shifts in response to environmental variability and climate change (e.g. IMMA designations include 10-year review periods to account for climate change-related shifts [97]).

Dynamic spatial management tools, which consider the shifting nature of the ocean and its users, can also be used to identify biologically important habitats [98]. An example of this flexible management approach is WhaleWatch, which uses location data (via Argos satellite transmitters) and remotely sensed environmental data (e.g. sea surface temperature, chlorophyll concentrations and wind speed) to predict where and when blue whales are likely to occur within the California Current [99]. Dynamic spatial management tools, such as WhaleWatch, are comparatively more robust to environmental variability and climate change because they use near real-time data to forecast marine mammal distributions. However, challenges include the amount of data, advanced analytical processing and modelling, and equipment maintenance required, and the need for improved data sharing and open access [98,100].

Additionally, the processes underpinning the distribution models may change, thus impacting their predictive ability [101]. Nevertheless, these tools can be used to inform activity planning and the extent of closures to disturbance-inducing activities or to enact real-time management actions [57,102].

## (b) Duration and frequency

The temporal features of disturbance also greatly influence the extent to which vital rates are impacted, as highlighted in several PCoD models [7,12,13,15,49,55]. For example, simulations conducted by New *et al.* [7] predicted that an increase in the number of disturbance days would lead to a decline in southern elephant seals' lipid mass and, subsequently, a decrease in pup weaning mass and survival. They also found that the predicted decrease in pup survival resulting from a prolonged disturbance (i.e. reducing the duration of a female's foraging trip by half) in any 1 year had seemingly minor impacts on the population. However, the effects of repeated exposures over a 30-year period led to a substantial decline in population size. Simulated disturbances for California sea lions yielded similar adverse effects on reproductive success, and, due to their income-breeding strategy, McHuron *et al.* [12] found that even relatively short, infrequent disturbances adversely affected population dynamics.

PCoD models demonstrate that reducing the duration and frequency of disturbances can help mitigate any energetic consequences by allowing individuals to regain energy reserves [13,15,49,55,57]. For example, Nabe-Nielsen *et al.* [49] found that varying the time and spatial distribution of wind farm construction in the North Sea, from ordered (where wind farms were built east to west) to random (where wind farms were built randomly), could reduce the duration of noise exposure experienced by harbour porpoises in important foraging areas. Additionally, they demonstrated that the length of breaks between piling events influenced the predicted effects of noise, where longer breaks gave individuals more time to recover. Farmer *et al.* [55] also found that less frequent disturbance provided opportunities for sperm whales to compensate for missed foraging, with the time to terminal starvation being roughly inversely proportional to the frequency of the disturbance event. Such information can support activity planning and area-specific caps on disturbance-inducing activities, especially within biologically important habitats [11,57].

## (c) Nature and context

Marine mammals are exposed to a wide variety of human activities, including offshore energy development, military training exercises, shipping, fishing and wildlife tourism [5]. In recent years, research on behavioural responses has largely focused on the potential vulnerability of marine mammals to acoustic disturbance sources (e.g. naval sonars, seismic airguns and vessels) [39,103], with individual responses often depending on the behaviour of the source (e.g. moving or stationary), the distance between the source and receiver, and the nature of the sound itself (e.g. source frequency and intensity) [94,104], which all affect the consequences of disturbance in PCoD models [57,96]. This research has led to the development of analytical tools such as dose–response functions, which provide a framework for relating an individual's probability of responding to some metric of exposure (e.g. received sound level) [104]. Several environmental factors (e.g. depth, bottom

sediments, sea ice and the sound speed profile of the water column) can also influence the propagation distance of the acoustic signal and thus the received sound level experienced by an individual [105].

An individual's propensity to respond and the severity of the response likely depend on additional, intrinsic factors. For example, PCoD models show that species [17], sex [55], age class [13] and body condition [61], as well as context (e.g. current behavioural state [96] or energetic state [32]), can influence the effect of disturbance on individual health and vital rates. An individual's experience can also influence the severity of response, although changes in responsiveness have yet to be incorporated into PCoD models. For example, a novel disturbance may cause an overt reaction, while prior experience may lead to habituation or sensitization [94,104].

Disturbance sources may have radically different effects depending on both intrinsic and extrinsic factors, making it difficult to compare across scenarios. Nevertheless, generalizations can be made (as demonstrated in this synthesis) regarding which factors have the greatest effects, and an improved understanding of the underlying processes for how and why individuals respond to disturbance may allow for more accurate predictions. As outputs of behavioural response studies become available, they should be incorporated into PCoD models and risk assessments (e.g. via improved dose–response functions) and used to develop mitigation and monitoring protocols that validate predictions [96,104].

# 4. Environmental conditions

Environmental conditions can influence female body condition and impact reproductive success via changes in prey, from distribution and abundance to composition and caloric value [106,107]. Changes in prey availability can occur across a range of spatial and temporal scales, from hour-long tidally driven hotspots and mesoscale features that can persist for months (e.g. eddies and fronts) to ocean basin-wide shifts resulting from within- (e.g. seasonal variability) and between-year (e.g. climatic oscillations) variability [108,109]. Climate change affects these natural oceanographic and atmospheric processes [110,111] and, subsequently, the distribution and viability of marine mammal populations [112,113]. In the following sections, we examine the importance of environmental conditions when assessing a population's sensitivity to disturbance and mediating risk (figure 1).

## (a) Natural variability in prey availability

PCoD models show that prey availability strongly influences a population's sensitivity to disturbance [13,15,91]. As a result, strategic planning for the timing of disturbance-inducing activities relies upon understanding the links between abiotic and biotic factors that drive marine mammal sensitivity to disturbance.

Within-year variability or seasonal changes in temperature and light level coupled with nutrient availability influence the distribution and abundance of primary producers, their consumers and higher trophic levels [114]. This results in annual cycles of high and low productivity, with higher latitudes experiencing more pronounced seasonal variation and higher ocean productivity than equatorial regions [115]. In response to seasonal variability, marine mammals have evolved various behavioural and life-history strategies to

maximize fitness. For example, prey availability has been shown to affect reproductive strategies, with higher food availability and seasonality favouring a capital-breeding strategy [51], which is reflected in the current distribution of many marine mammal populations [116].

Capital breeders, including many mysticetes and phocids, cycle between periods of intensive foraging and fasting that are synchronized with seasonal changes in productivity [117]. Because there is a limited period to acquire energy, PCoD models show that disturbance-inducing activities that reduce foraging time can affect an individual's energy balance and thus reproduction and survival [11,13]. However, the magnitude of the effect will likely depend on the proximity of the disturbance source to important foraging areas, as shown in some PCoD models [13,61], and whether the disturbance coincides with periods of increased energy intake [118]. By contrast, income breeders, including many odontocetes and otariids, rely on more predictable environments to finance their expensive lifestyle. In response to seasonal variability, many will shift their diet, change their foraging behaviour and/or travel to more productive areas [119]. However, PCoD models show that some populations may be more spatially and/or temporally restricted in their ability to adapt to disturbance-induced changes in foraging during periods of low prey availability [15] and increased energy intake [32]. As such, this has important implications for the timing of disturbance-inducing activities, particularly where prey or the environment fluctuates seasonally.

Between-year variability, including climatic oscillations, also affects prey availability by exerting large control over physical (e.g. intensity and direction of ocean currents) and biological (e.g. intensity and duration of bottom-up productivity) processes [109]. Some climatic oscillations follow predictable cycles or quasi-predictable patterns, while others have no periodicity [109]. One of the strongest modes of climate variability is the El Niño Southern Oscillation (ENSO), which causes increased sea surface temperature and decreased upwelling and biological productivity for up to a year on a 3- to 7-year cycle [120]. Although ENSO originates in the eastern tropical Pacific Ocean, its impacts can be felt at higher latitudes [106], and its effects can be amplified significantly by coinciding with other modes of climate variability (e.g. Pacific Decadal Oscillation (PDO) [121]). Dramatic impacts on the vital rates and distribution of marine mammals due to changes in productivity have been described for previous ENSO events [75] and other climatic oscillations, such as the North Atlantic Oscillation [122], PDO [123] and Southern Annular Mode [124]. A disturbance-inducing activity that overlaps with ENSO or a similar large-scale anomaly (including marine heatwaves [125]) will likely interact with the effects of these naturally occurring events. For example, Pirotta et al. [13] found that, during an ENSO event, the location, duration and frequency of a simulated disturbance mediated the cumulative effect on blue whale vital rates.

PCoD models show that changes in prey availability interact with disturbance sources to affect reproductive success and survival [13,15,91], which emphasizes the importance of timing to minimize risk when planning a disturbance-inducing activity. The outputs of PCoD models can be used to determine whether an activity can be carried out in a way that does not exacerbate the energetic impacts of local or regional environmental conditions. Model parametrization, however, will depend on the wealth of knowledge about

environmental conditions. For example, the patterns that drive biophysical change on seasonal timescales are well studied and can be used to forecast prey and marine mammal hotspots [99,126]. However, the underlying forces driving climatic oscillations are less understood and more unpredictable (see table 1 in [109] for the periodicity, distribution and characteristics of the primary climatic oscillations in the ocean). Nevertheless, identifying the climatic oscillations in the disturbed area, including the periodicity and most recent occurrence, can help wildlife managers, practitioners and industry understand the probability of a proposed activity overlapping with these periods of reduced prey availability.

## (b) Climate change

Aquatic and terrestrial ecosystems are undergoing complex changes in response to climate change. Observed and predicted changes include warmer temperatures, higher sea levels, increased acidity and reduced sea ice, as well as changes in precipitation patterns and the frequency of storms and extreme events [127]. For marine mammals, these changes may cause geographical range shifts, loss of habitat and changes in the food web, as well as increased exposure to toxins and susceptibility to disease [112,113]. Climate change may also result in new and more frequent interactions between marine mammals and humans [128]. Therefore, climate change coupled with anthropogenic disturbance may have interactive and cumulative effects on reproductive success and survival [129,130].

While several PCoD models have examined how changes in prey availability (via natural variation or anthropogenic disturbance) affect individual vital rates and population dynamics, they have only recently started to assess the effects of disturbance occurring within a changing environment. Pirotta *et al.* [13] found that, in the absence of disturbance, simulated scenarios of environmental change (using changes in the frequency of ENSO and declines in productivity as proxies) could have severe consequences on the vital rates of eastern North Pacific blue whales, with females tolerating only small reductions in overall productivity and prioritizing their survival at the expense of reproduction. In scenarios with both environmental change and anthropogenic disturbance, they found that the combined effects on vital rates might be larger than in isolation.

The evolutionary history of marine mammals demonstrates their ability to adapt to ecosystem change, but long generation times restrict the rate at which evolution can occur [113]. While highly mobile populations may be able to respond more rapidly to climate change through phenotypic plasticity, others may be less able to adapt and thus more sensitive to changing conditions, including populations that have specialized diets, occupy reduced or fragmented geographical ranges or depend on specific substrates or sites for important life-history stages (e.g. sea ice for pupping) [112,113].

A population's vulnerability to climate change will probably be determined by the magnitude of change expected to occur within its current distribution, as well as its sensitivity to environmental change and adaptive capacity to respond [131]. Climate vulnerability assessments can help identify which populations may be most vulnerable to climate change and thus more at risk to a proposed activity under changing environmental conditions, especially for activities that span several years to decades, but these assessments will remain challenging for most populations.

## 5. Data gaps and future priorities

The real-world applications of the PCoD framework have provided valuable insights into implementation challenges and data gaps, thereby guiding model development and focusing data collection (also see [2]).

The amount of data and processing time required for PCoD models can limit their direct application in decisions about proposed activities. Additionally, due to species and contextual differences, there is not a simple, 'one-size-fits-all' approach for applying the PCoD framework in risk assessments. As such, its application will need to be adapted based on the data available and issue being addressed. Collaborations between modellers and wildlife managers can help identify ways to increase model accessibility and adaptability and the outputs necessary to make decisions. Furthermore, model assumptions and their influence on outputs should be made transparent to and considered by end users, such as wildlife managers and policymakers, in subsequent decisions (see the electronic supplementary material, *PCoD model assumptions and limitations*). Critical data gaps for the parametrization of PCoD models include prey availability across a population's range, including abundance, variability, type and energetic content; disturbance-related changes in physiology that compromise individual health; baseline health dynamics and how they relate to variation in vital rates and the effect of intrinsic and extrinsic contextual factors on an individual's response to disturbance. Efforts are currently underway to systematically evaluate these gaps and inform future research to better parametrize PCoD models. In the absence of sufficient empirical data, an interim PCoD approach has been used in risk assessments and parametrized via expert elicitation to quantify the relationship between changes in behaviour and physiology to fitness [19,132,133].

Extensive baseline knowledge on demography is also required, including life-history parameters (e.g. population growth rate and age at first reproduction or sexual maturity) and abundance estimates [2]. However, this information is not known for many marine mammal populations (but see [134] for how monitoring programmes can be designed to collect such data), which makes quantitative population-level analyses difficult to undertake. In the absence of demographic information for the target population or a related species, first principles can be used to predict how the population may respond to disturbance [135]. For example, Nattrass & Lusseau [135] demonstrate how a basic understanding of a species' physiology and the productivity dynamics of the environment can be used to estimate a population's resilience to disturbance. Many of the general principles explored in this synthesis can help inform such an assessment.

To determine whether the effects of disturbance, as predicted by PCoD models, are compatible with the conservation objective for a given population or stock, allowable harm limits also need to be established [136]. This has important implications for determining whether a disturbance-inducing activity can move forward as proposed or whether mitigation strategies are needed to reduce potential effects. For example, the United States' MMPA's Potential Biological Removal (PBR) equation represents the number of individuals that can be removed annually (not including natural mortalities) while allowing the population to reach or maintain its optimum sustainable population size [137]. While PBR only accounts for the cumulative effects of severe and lethal injuries by commercial

fisheries, it has the potential to include non-fishery mortalities and the cumulative impact of sublethal effects. For example, the National Research Council [8] provided recommendations for how PBR can be improved to reflect total mortality losses and other cumulative impacts, including assigning severity scores to various physical injury and behavioural harassment levels. Expanding PBR to include sublethal effects is just one example of how allowable harm limits can be established.

Gauging allowable harm limits for a proposed activity requires wildlife managers to consider the cumulative effects of multiple disturbance sources [136]. As such, the PCoD framework has recently been extended to include the effects of multiple disturbance sources, or stressors, that may cumulate [138]. Research into the Population Consequences of Multiple Stressors (PCoMS) framework is just beginning, and several key research needs include identifying the stressors and dosages to which individuals are exposed, determining dose–response functions to predict the effects of single stressors and understanding the mechanistic pathways underpinning the effects of varying stressor combinations [138].

PCoD models can provide valuable insights into which contextual factors influence a population's degree of exposure and sensitivity to disturbance, including species for which data are limited and models are unavailable. By identifying emerging themes in existing PCoD models, we have highlighted general principles to consider when assessing risk. Future models could be developed for representative populations or species exposed to common disturbance scenarios to investigate broad patterns in population responses to disturbance (e.g. see [31,92]). By identifying population characteristics and other contextual factors that could lead to population-level effects, model findings could be used to further guide decision-making and develop mitigations that target populations most at risk or sensitive to a proposed activity. Ultimately, advancing the PCoD and PCoMS frameworks will provide wildlife managers, practitioners and industry with the information necessary to better assess and effectively mitigate disturbance risks so that environmental and social considerations can be balanced.

Data accessibility. This article has no additional data.

Authors' contributions. K.A.K. and D.P.C. conceived the manuscript, K.A.K. led the writing of the manuscript, and K.A.K., R.S.B., E.P. and D.P.C. contributed critically to the drafts and revisions of the manuscript. All authors read and approved the final manuscript.

Competing interests. We declare we have no competing interests.

Funding. K.A.K., R.S.B. and D.P.C. were supported by the E&P Sound and Marine Life Joint Industry Programme (JIP) of the International Association of Oil and Gas Producers (IOGP) (grant no. 00-07-23). K.A.K. was also supported by the National Defense Science and Engineering Graduate (NDSEG) Fellowship. E.P. was supported by the Office of Naval Research (ONR) (grant no. N00014-19-1-2464). D.P.C. was also supported by ONR (grant no. N00014-18-1-2822).

Acknowledgements. We thank the IOGP JIP, ONR and NDSEG Fellowship for providing funding for the synthesis and their continued support of PCoD efforts, and the PCoD community for their continued contributions to the literature and advancement of the field. We also thank members of the Costa Lab for feedback on the manuscript figures, Rachel Holser for helpful discussions about global oceanographic processes and Stephanie Adamczak for feedback on the manuscript text. Finally, we thank the associate editor and three anonymous reviewers for their comments on the manuscript.

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
