## [Peer Review File · Proceedings of the Royal Society B: Biological Sciences]

Review History

RSPB-2021-0325.R0 (Original submission)

Review form: Reviewer 1

Recommendation

Accept with minor revision (please list in comments)

Scientific importance: Is the manuscript an original and important contribution to its field?

Excellent

General interest: Is the paper of sufficient general interest?

Excellent

Quality of the paper: Is the overall quality of the paper suitable?

Excellent

Is the length of the paper justified?

Yes

Should the paper be seen by a specialist statistical reviewer?

No

Do you have any concerns about statistical analyses in this paper? If so, please specify them explicitly in your report.

No

It is a condition of publication that authors make their supporting data, code and materials available - either as supplementary material or hosted in an external repository. Please rate, if applicable, the supporting data on the following criteria.

Is it accessible?

N/A

Is it clear?

N/A

Is it adequate?

N/A

Do you have any ethical concerns with this paper?

No

Comments to the Author

This is a well written review paper that describe and discuss an important type of models. In this synthesis the authors reviewed common themes about Population Consequences of Disturbance (PCoD) models. They highlighted essential factors that need to be considered in this kind of models and identified data gaps and suggested future directions. To be honest, I am not an expert in this kind of models and thus would not be able to provide in-depth comments. I do get very excited about this kind of models by reading this manuscript, and believe this article will be very useful for many people like myself to learn more about this kind of models. Therefore, from my point of view I would love to see this article published in a high-level journal such as Proceedings B. I nonetheless have a few very minor comments for the authors to consider.

Line 83-84. "... although we focus on marine mammals, these general principles could guide risk assessments for other wildlife species." I understand that the authors have focused on marine mammals (which is totally fine). I am just curious if this kind of models have been applied to other taxa (e.g., terrestrial large mammals). I think it will be great if the authors could provide just a few examples (citations) for applications in taxa other than marine mammals. If such examples do not exist, what would be the potential challenges to apply this kind of models to other taxa? It will be great if the authors could briefly discuss this kind of issues (and my apologies if the authors have already done so but I overlooked it).

It seems that this kind of models can have flexible structures and can incorporate multiple types of information. Do these models have a common overall structure or even a common mathematical form? If they could take a common overall structure, it would be great if the authors could provide a figure to illustrate that. I appreciate the nice figures the authors provided, but it seems there lacks something that can guide the readers to start developing such a model.

I again believe that the manuscript is well organized, and Figure 1 in particular summarize and help the readers to understand the main points very well. I just wonder if there are other factors that need to be considered in the "Environmental Conditions" section. In particular, I would think that human disturbances such as hunting/poaching is still a major threat to marine mammals. Other things I can think about include pollution (e.g., plastics) and collision between animals and ocean vessels. Can these things be considered in this kind of models? If so, shall these be discussed in addition to prey abundance and climate change? I think discussing these things are important as terrestrial animals may face similar issues (e.g., poaching, traffic).

Review form: Reviewer 2

Recommendation

Accept with minor revision (please list in comments)

Scientific importance: Is the manuscript an original and important contribution to its field?

Excellent

General interest: Is the paper of sufficient general interest?

Good

Quality of the paper: Is the overall quality of the paper suitable?

Good

Is the length of the paper justified?

Yes

Should the paper be seen by a specialist statistical reviewer?

No

Do you have any concerns about statistical analyses in this paper? If so, please specify them explicitly in your report.

No

It is a condition of publication that authors make their supporting data, code and materials available - either as supplementary material or hosted in an external repository. Please rate, if applicable, the supporting data on the following criteria.

Is it accessible?

Yes

Is it clear?

Yes

Is it adequate?

Yes

Do you have any ethical concerns with this paper?

No

Comments to the Author

General Comments

Because the fundamental approach uses bioenergetic models as the basis of PCOD, I would like to see some more explicit description of whether and how such a perspective has been applied in terms of describing and predicting population demographics in other taxa. I know this has been done in non-marine mammals and some examples could help substantiate the use of bioenergetics to populations within the context of disturbance here

My main suggestion relates to the need to specifically consider and honestly discuss the underlying prior assumptions, which have been well or poorly parameterized by data, and what the limitations of previous PCOD assessments are. One of the primary criticisms of and limitations to how much impact these modeling approaches have actually had in management decisions is that they are not sufficiently transparent in their assumptions and which are strongly or weakly supported and that the underlying mathematical processes are not clear and replicable

for non-specialists (like decision makers). Is this a fair criticism? I kind of think it is to be honest but the authors may not. Either way, I'd strongly suggest this be directly and explicitly addressed in the intro and then especially picked up in data gaps and future priorities. The parameterization discussion is there and I do like the discussion of limitations of PBR (even though it is quite US centric). But the reader is left wanting at the end in my opinion about how best to push these fantastic conclusions from the emerging themes (I love Fig 1 - I'm going to put it on my wall) really and practically into play. Specifically, how will advancing PCoD and PCoMS help managers - last sentence? How specifically do you think these tools should be applied - risk assessment frameworks, conceptual models, within biological opinions as a reference? We can't expect a resource analysts at NMFS or BOEM or JNCC to be able to run or even fully understand the details of these complex models. This is an impediment to their direct application. While I think this paper goes a long, long way into extracting the key points (again Fig 1) but I just encourage you to specifically suggest how best people in those scenarios that aren't versed in the details of the stats should really try and actually apply something like the recent PDoD results in a practical scenario.

Specific comments

Abstract- lines 13-15. The second part of this sentence is really important and may not be inherently familiar in terms of what you mean to readers of PRSB. I suggest you make this a separate and simple sentence and emphasize the important role that such quantitative models can play in better parameterizing probabilistic risk assessments for assessing the longer-term severity of disturbance.

Line 19. I would include 'assumptions' and 'limitations' around the word 'findings' (see general comments on the need for this in my view)

Abstract. Somewhere in the second half of the abstract I think the term 'transparent' or 'evident to non-specialists' or non-statisticians should appear. One of the main limitations I have seen in practical applications is that the priors and assumptions are not readily evident and transparent (also addressed above).

Abstract. There is no indication of taxa emphasis within the abstract - the term marine mammals is not used but should be woven in somewhere and briefly why this is the case.

Intro - first paragraph. Would like to see a more recent reference than NRC 2005 and many more updated broad reviews. Additionally, later in the paragraph, a good recent reference spanning national and international policy implications of this issue is: Chou, E., et al. (2021). International policy, recommendations, actions and mitigation efforts of anthropogenic underwater noise. *Ocean & Coastal Management*, 202, 105427.

Line 53. Very good to mention whale watching here. In case any other reviews balk at this, I believe it is a very apt inclusion. Could arguably add elevated background noise in high traffic areas as well

Figure 1. Fantastic. This is the most important and impactful set of messages in the paper.

Line. 94 - Section on movement ecology. Lot of good and specific references here. One important one that reviewed this topic across a number of taxa and made a very clear point earlier along these lines is Forney et al. 2017. This could be used instead of several of the many references here (there are quite a lot of refs in the paper overall and if anything this section is more detailed than I thin kit needs to be), but it definitely should appear here somewhere as it was one of the first papers to make this overall pointy about movement and susceptibility to disturbance. Forney, K. A., et al. (2017). Nowhere to go: noise impact assessments for marine mammal populations with high site fidelity. *Endangered Species Research*, 32, 391-413.

p. 178 Section on Reproductive strategies. This section is better focused and tighter in the points made than the one above on movement. I suggest mirroring the level of detail and focus to that in this section

Lines 408-413. Nature and context. In contrast, this section is really light on something that is fundamentally and even overarchingly important at least in terms of acute responses. There should be some discussion of novelty and habituation here and a more robust consideration of factors like behavioral state as a contextual factor influencing the type and severity of response. I also think that the section below seems to paint a bit of a hopeless scenario about potential generalization across scenarios. Just because there is context-dependency it doesn't mean we can't make reasonable and transparent kinds of assumptions about context scenarios - as done in Pirotta et al. (2021) ref 108. This should be specifically noted here - the paper is referenced but this is a key point.

Section 5 is pretty light on some key points - addressed above

Review form: Reviewer 3

Recommendation

Accept with minor revision (please list in comments)

Scientific importance: Is the manuscript an original and important contribution to its field?

Good

General interest: Is the paper of sufficient general interest?

Good

Quality of the paper: Is the overall quality of the paper suitable?

Excellent

Is the length of the paper justified?

Yes

Should the paper be seen by a specialist statistical reviewer?

No

Do you have any concerns about statistical analyses in this paper? If so, please specify them explicitly in your report.

No

It is a condition of publication that authors make their supporting data, code and materials available - either as supplementary material or hosted in an external repository. Please rate, if applicable, the supporting data on the following criteria.

Is it accessible?

N/A

Is it clear?

N/A

Is it adequate?

N/A

Do you have any ethical concerns with this paper?

No

Comments to the Author

General comments:

In this interesting synthesis, Keen et al. review existing Population Consequences of Disturbance models for marine mammal species to identify emerging themes in model predictions. The review presents the principles of PCoD models very well and is a timely contribution to the wider literature as PCoD models become more popular. The paper is generally well written, and the figures are excellent.

I do, however, think that from the beginning it should be clear that, while the emerging themes may also be relevant for other species, this review is focused entirely on marine mammal science and models. While the PCoD title has been primarily used for marine mammal models, an expansive suite of models exist which focus on the impacts of disturbance on non marine mammal wildlife populations. I think that highlighting this as a marine mammal centered review from the beginning ("marine mammal" is not currently mentioned in the title or abstract) could be advantageous to avoid suggesting that the review will focus on models of population impacts of disturbance at large. If the claim is made that the findings presented from marine mammals are generally transferable, the review would benefit from providing support of this claim throughout the text using either empirical or model findings from other species. It would be particularly interesting to compare results from animals with similar life history patterns but with substantial differences in other regards, e.g., large whales and elephants (Boult et al. 2019 - doi.org/10.1111/csp2.87). Though I think that the focus on acoustic disturbance in PCoD models could complicate comparisons and that it would take a considerable amount of effort to push the article towards being more inclusive of non marine mammal species. Considering this, if sticking with marine mammals, I would be a bit more tentative in making the claim that the findings for these specific marine mammal models can be used to predict risk in other species. While concepts presented may be generally useful, such as lactation being an energetically expensive and risky period for income breeders, I think that it is important to stress that without a full consideration of species-specific behavior and physiology and the environment it will be challenging to make accurate predictions of the population-level responses to disturbance.

Throughout the text both empirical results and model findings are presented. I think as a synthesis on the results of PCoD models, it needs to abundantly clear what information is coming from what source. This is well done in the reproductive strategies section but I believe that other sections could benefit from increased clarification.

With these relatively minor changes, I believe the authors can make the paper more accessible and useful to the wider wildlife modelling and conservation community.

Detailed comments:

L40-42: Have PCoD models been used in impact assessments? Would be interesting to note if so.

L81-82: Do we know that the findings of existing models are generally true? The complicated nature of disturbance responses may make it challenging to directly apply the findings from a model developed for one species to another even if life history patterns are similar.

L85, 334, & 426: Maybe these sections could just be titled "Life-history traits", "Disturbance source characteristics", etc.? I don't think the "The importance of..." part is necessary.

L116: It would be nice to have some provided examples of these lasting effects.

L117-118 & 126: Maybe it would be worth combining these two statements about some individuals in migrating populations not migrating?

L142-143: Could add a "single" disturbed area to highlight

L168-177: Many PCoD models do not explicitly consider animal movement in their simulations. It would be nice to briefly describe to what extent (and under which scales) has movement actually been considered in PCoD models and what scales of movement are important for measuring

disturbance responses.

L187: Could be good to specify such as “These sensitivities to disturbance in income breeders...”

L188-189: Is this statement supposed to be related to the lactation period? As all marine mammals can store lipids for later use to some degree, is this intended to state that capital breeders may be less sensitive to foraging losses during the lactation period as lipid used for this process has already been stored? Would be good to clarify.

L200: Could add that this could ultimately impact population abundance.

L204-207: But it should be highlighted that we know very little about the specifics of how and when these decisions are made in marine mammals, particularly cetaceans.

L231-232: When known? These predictions may be very tough to get for many species/environments.

L236: Instead of basal metabolism I would say survival as it seems that this here includes costs additional to true basal metabolism, including activity, thermoregulation, feeding, etc.

L243: they require a “relatively” higher resource acquisition rate, but not necessarily absolutely unless this rate is per unit time and mass.

L256-258: Though small animals need relatively more food per unit mass they also require less food in total, for periods when food is limited smaller animals may more easily be capable of meeting their energetic needs. There are many benefits to being small which should also be discussed in this section. See:

Goldbogen, J.A., Cade, D.E., Wisniewska, D.M., Potvin, J., Segre, P.S., Savoca, M.S., Hazen, E.L., Czapanskiy, M.F., Kahane-Rapport, S.R., DeRuiter, S.L. and Gero, S., 2019. Why whales are big but not bigger: Physiological drivers and ecological limits in the age of ocean giants. *Science*, 366(6471), pp.1367-1372.

L271-273: They also have additional costs of growth.

L276: Maybe “may be exposed” rather than just “present”?

L311: Would be nice to reference Read & Hohn 1995 here.

L315: Citation needed?

L315-317: Can cite Nabe-Nielsen et al. 2018 here as the bounce back after disturbance is visible in their porpoise simulations.

L325-330: This feels a bit tagged on the end, also if this is tied to body condition estimates, I would assume that this sort of pattern would still come out of many of the cited models, but not if it is related to differences in behavior, genetics, etc.

L329-330: What would be the population level implications of this finding?

L337: Nature is a bit vague, maybe it would be good to provide an example or short description here.

L345-348: This sentence could be reworded to be more concise.

L353: Also important that the similar habitat is of a sufficient area.

L365: "consider"

L368-381: How can these technologies be paired with PCoD models?

L384-390: These two sentences have quite a few subordinate clauses in the middle of the sentence. Some rewording could make for smoother reading.

L404-405: Second “inform” isn’t needed.

L407: How are disturbances modelled to impact individuals in these different models (e.g., halting energy intake)? What are the common approaches?

L416-420: This phrasing implies that received level doesn’t factor at all into responses.

L441: “sensitivity to disturbance”.

L451-454: What about seasonal or temporal variations in energy intake needs which are not related to reproduction but instead the environment? e.g. the seasonality in energy balance presented in:

Gallagher, C.A., Grimm, V., Kyhn, L.A., Kinze, C.C. and Nabe-Nielsen, J., 2021. Movement and seasonal energetics mediate vulnerability to disturbance in marine mammal populations. *The American Naturalist*, 197(3), pp.296-311.

L479-481: Just influences? Would be nice to be more specific here.

L514: Were these combined effects synergistic?

L530: This section could be a bit expanded a bit to be more specific about identified data gaps across the different modelling exercises. It would be interesting to see for models in which sensitivity analyses were carried out, if there were any common themes identified for types of data that these sorts of models are particularly sensitive to.

L541-542: I would say that for the vast majority of populations this information isn't known, what are the implications of that? and how do we get around this issue?

Figure 1: It is a little difficult to read the colored text (particularly the green). I would use bold to make this more legible for those of us with inferior vision.

Figure 2a: I find it a bit challenging to read this figure. I think that, as is, it doesn't communicate the main points effectively. In the Nomadic and Migratory populations it is unclear whether tracks are from a single or multiple individuals. In the original plot in Costa et al. 2016 the individual tracks were color coded so it was clear, though in the current plot it is already difficult to tell the difference between the pink and orange color at the current size. Is it necessary to have two disturbance zones? It could be nice to have a single zone shown, maybe as a fill with high transparency, and have the individual tracks colored as in the original figure to make clear the different number of individuals being disturbed.

Figure 3: It would be nice to include variability in this visual as well around the start and end of each period to encourage the consideration of peak periods if relevant - as in the Cornell birds of the world annual cycles. Also I'm not sure that having maintenance here helps anything since it is visualized as constant year round.

Decision letter (RSPB-2021-0325.R0)

26-Apr-2021

Dear Ms Keen:

Your manuscript has now been peer reviewed and the reviews have been assessed by an Associate Editor. The reviewers' comments (not including confidential comments to the Editor) and the comments from the Associate Editor are included at the end of this email for your reference. As you will see, the reviewers and the Editors have raised some concerns with your manuscript and we would like to invite you to revise your manuscript to address them.

We do not allow multiple rounds of revision so we urge you to make every effort to fully address all of the comments at this stage. If deemed necessary by the Associate Editor, your manuscript

will be sent back to one or more of the original reviewers for assessment. If the original reviewers are not available we may invite new reviewers. Please note that we cannot guarantee eventual acceptance of your manuscript at this stage.

Research ethics:

Use of animals and field studies:

It is a condition of publication that you make available the data and research materials supporting the results in the article. Please see our Data Sharing Policies (<https://royalsociety.org/journals/authors/author-guidelines/#data>). Datasets should be deposited in an appropriate publicly available repository and details of the associated accession number, link or DOI to the datasets must be included in the Data Accessibility section of the article (<https://royalsociety.org/journals/ethics-policies/data-sharing-mining/>). Reference(s) to datasets should also be included in the reference list of the article with DOIs (where available).

Please submit a copy of your revised paper within three weeks. If we do not hear from you within this time your manuscript will be rejected. If you are unable to meet this deadline please let us know as soon as possible, as we may be able to grant a short extension.

Best wishes,
The Proceedings B Team
mailto:proceedingsb@royalsociety.org

Associate Editor
Comments to Author:

Thank you for submitting your interesting manuscript for consideration as a PRSB Evidence Synthesis article. I would like to start out by apologising the time it has taken to secure appropriately high quality and representative referee reports, but as you will see below, despite the delay, I hope that you find the level of detail and constructive tone of significant help in taking your work forward.

Your work has now been reviewed by 3 experts in the field, and I have read the manuscript myself. While there is a consensus that the topic is pertinent for an evidence synthesis article, and timely in that it is likely to command broad interest, you will see below some detailed and constructive suggestions of how to improve your MS further. You will see that a range of suggestions indicating collectively that your manuscript and topic, is in my opinion, worthy of additional investment and consideration, and accordingly invite a revision.

As such, I encourage you strongly to consider the comments below, and submit a revised manuscript at your earliest convenience. You will see that there are a variety of concerns which I echo, and highlight here, though full details are provided in the respective referee reports. A fundamental concern cross referees, which I endorse, is the extent to which your consideration can be applied across non-marine mammal taxa. I appreciate that this will require additional text, though indicating the extent to which such an approach is transferable, with brief illustrations of some examples, or indeed, how and why generic value is constrained, would be of profound interest to the wide readership of PRSB. To what extent has this kind of model been applied to other taxa? Some sharpening of the justification for underlying assumptions of the approach is also required, as well as the extent to which these types of models may, or may not, have a common overall structure or mathematical form. The latter may be most effectively summarised in a Figure, as proposed below, alongside other existing highly informative schematics. Enhanced accessibility to our readership of your approach and applications, would be enhanced by such additional information, without necessarily increasing the length significantly.

It is also important to provide a more robust and justified rationale for the choice of factors that need to be considered, especially in the section of environmental conditions, with some specific suggested factors given below. Indeed, as one of the referees highlights, is a major suggestion, a more substantive critique on the underlying prior assumptions which have been well or poorly parameterised by data, including previous limitations of PCOD assessments would be of significant benefit. It is of course vitally important in evidence synthesis articles, wherever possible, to emphasise practical utility and translation of empirical data/evidence, into practical policy and in this case, facilitating managers in the context of risk assessment frameworks. One final general point, initial to the numerous helpful suggestions, is the methodology employed in accessing and synthesising the evidence base presented. It is especially vital within evidence synthesis manuscripts to provide a robust, transparent and representative framework with explicit clarity on the source and choice of information presented. While this is relatively clear in the section on reproductive strategies, it is perceived to be less so elsewhere.

Much of the content of the constructive referee reports is self-evident, and I hope should you decide to revise your manuscript, will provide a useful guide of how to proceed. I would like to additionally draw your attention specifically to our requirements for publication of Evidence Synthesis articles. Notwithstanding, in your response to referees, I would be grateful if you would include a brief account relating to Editorial comments, on how the manuscript has been modified in relation to my brief suggestions. In particular, as you will have seen from the guidelines available for our Evidence Synthesis articles (<https://royalsocietypublishing.org/rspb/evidence-synthesis>), it is vital that the reader is able to assess the validity, robustness and objectivity of the evidence base presented. Importantly also, when putting the final touches to the article, please ensure wherever possible, that where relevant, you have addressed some of the questions below, that characterises the Evidence Synthesis article type, though I fully recognise, that many questions will only partially apply to your manuscript :

1. Is the key policy-related question(s) articulated clearly?
2. Is there a clear justification in support of policy relevance?
3. Is the likely target audience identified clearly?
4. Does the search for literature utilise a comprehensive range of sources?
5. Does the synthesis article apply clearly documented inclusion criteria to all potentially relevant studies found during the search?
6. Is a clear methodology described to avoid bias?
7. Is your study objectively weighted according to methodological quality of cited literature?
8. Are knowledge gaps and priorities clearly identified?
9. Are outcomes/recommendations tangible in terms of likely impact?
10. Are all necessary supporting information available and accessible??

Including a brief indication of how you have addressed the specific criteria above in your response letter would be most helpful. I appreciate that the volume of revision is extensive, and may go beyond what you had originally anticipated. Notwithstanding, I would hope you will find the constructive and detailed suggestions helpful in formulating a more robust and representative evidence synthesis article for resubmission. As indicated below, as in all peer review processes, the invitation to revise, is of course no guarantee of eventual publication, but I will do my best to exercise consistency in the remaining peer review process, by approaching the original referees, at a minimum, though of course I am not in a position to confirm their availability.

Thank you in advance for bringing this information together, and we look forward to receiving the revised manuscript in due course.

Reviewer(s)' Comments to Author:

Referee: 1

Comments to the Author(s)

This is a well written review paper that describe and discuss an important type of models. In this synthesis the authors reviewed common themes about Population Consequences of Disturbance (PCoD) models. They highlighted essential factors that need to be considered in this kind of models and identified data gaps and suggested future directions. To be honest, I am not an expert in this kind of models and thus would not be able to provide in-depth comments. I do get very excited about this kind of models by reading this manuscript, and believe this article will be very useful for many people like myself to learn more about this kind of models. Therefore, from my point of view I would love to see this article published in a high-level journal such as Proceedings B. I nonetheless have a few very minor comments for the authors to consider.

Line 83-84. "... although we focus on marine mammals, these general principles could guide risk assessments for other wildlife species." I understand that the authors have focused on marine mammals (which is totally fine). I am just curious if this kind of models have been applied to other taxa (e.g., terrestrial large mammals). I think it will be great if the authors could provide just a few examples (citations) for applications in taxa other than marine mammals. If such examples do not exist, what would be the potential challenges to apply this kind of models to other taxa? It will be great if the authors could briefly discuss this kind of issues (and my apologies if the authors have already done so but I overlooked it).

It seems that this kind of models can have flexible structures and can incorporate multiple types of information. Do these models have a common overall structure or even a common mathematical form? If they could take a common overall structure, it would be great if the authors could provide a figure to illustrate that. I appreciate the nice figures the authors provided, but it seems there lacks something that can guide the readers to start developing such a model.

I again believe that the manuscript is well organized, and Figure 1 in particular summarize and help the readers to understand the main points very well. I just wonder if there are other factors that need to be considered in the "Environmental Conditions" section. In particular, I would think that human disturbances such as hunting/poaching is still a major threat to marine mammals. Other things I can think about include pollution (e.g., plastics) and collision between animals and ocean vessels. Can these things be considered in this kind of models? If so, shall these be discussed in addition to prey abundance and climate change? I think discussing these things are important as terrestrial animals may face similar issues (e.g., poaching, traffic).

Referee: 2

Comments to the Author(s)

General Comments

Because the fundamental approach uses bioenergetic models as the basis of PCOD, I would like to see some more explicit description of whether and how such a perspective has been applied in terms of describing and predicting population demographics in other taxa. I know this has been done in non-marine mammals and some examples could help substantiate the use of bioenergetics to populations within the context of disturbance here

My main suggestion relates to the need to specifically consider and honestly discuss the underlying prior assumptions, which have been well or poorly parameterized by data, and what the limitations of previous PCOD assessments are. One of the primary criticisms of and limitations to how much impact these modeling approaches have actually had in management decisions is that they are not sufficiently transparent in their assumptions and which are strongly or weakly supported and that the underlying mathematical processes are not clear and replicable for non-specialists (like decision makers). Is this a fair criticism? I kind of think it is to be honest

but the authors may not. Either way, I'd strongly suggest this be directly and explicitly addressed in the intro and then especially picked up in data gaps and future priorities. The parameterization discussion is there and I do like the discussion of limitations of PBR (even though it is quite US centric). But the reader is left wanting at the end in my opinion about how best to push these fantastic conclusions from the emerging themes (I love Fig 1 - I'm going to put it on my wall) really and practically into play. Specifically, how will advancing PCoD and PCoMS help managers - last sentence? How specifically do you think these tools should be applied - risk assessment frameworks, conceptual models, within biological opinions as a reference? We can't expect a resource analysts at NMFS or BOEM or JNCC to be able to run or even fully understand the details of these complex models. This is an impediment to their direct application. While I think this paper goes a long, long way into extracting the key points (again Fig 1) but I just encourage you to specifically suggest how best people in those scenarios that aren't versed in the details of the stats should really try and actually apply something like the recent PDoD results in a practical scenario.

Specific comments

Abstract- lines 13-15. The second part of this sentence is really important and may not be inherently familiar in terms of what you mean to readers of PRSB. I suggest you make this a separate and simple sentence and emphasize the important role that such quantitative models can play in better parameterizing probabilistic risk assessments for assessing the longer-term severity of disturbance.

Line 19. I would include 'assumptions' and 'limitations' around the word 'findings' (see general comments on the need for this in my view

Abstract. Somewhere in the second half of the abstract I think the term 'transparent' or 'evident to non-specialists' or non-statisticians should appear. One of the main limitations I have seen in practical applications is that the priors and assumptions are not readily evident and transparent (also addressed above).

Abstract. There is no indication of taxa emphasis within the abstract - the term marine mammals is not used but should be woven in somewhere and briefly why this is the case.

Intro - first paragraph. Would like to see a more recent reference than NRC 2005 and many more updated broad reviews. Additionally, later in the paragraph, a good recent reference spanning national and international policy implications of this issue is: Chou, E., et al. (2021). International policy, recommendations, actions and mitigation efforts of anthropogenic underwater noise. *Ocean & Coastal Management*, 202, 105427.

Line 53. Very good to mention whale watching here. In case any other reviews balk at this, I believe it is a very apt inclusion. Could arguably add elevated background noise in high traffic areas as well

Figure 1. Fantastic. This is the most important and impactful set of messages in the paper.

Line. 94 - Section on movement ecology. Lot of good and specific references here. One important one that reviewed this topic across a number of taxa and made a very clear point earlier along these lines is Forney et al. 2017. This could be used instead of several of the many references here (there are quite a lot of refs in the paper overall and if anything this section is more detailed than I thin kit needs to be), but it definitely should appear here somewhere as it was one of the first papers to make this overall pointy about movement and susceptibility to disturbance. Forney, K. A., et al. (2017). Nowhere to go: noise impact assessments for marine mammal populations with high site fidelity. *Endangered Species Research*, 32, 391-413.

p. 178 Section on Reproductive strategies. This section is better focused and tighter in the points made than the one above on movement. I suggest mirroring the level of detail and focus to that in this section

Lines 408-413. Nature and context. In contrast, this section is really light on something that is fundamentally and even overarchingly important at least in terms of acute responses. There should be some discussion of novelty and habituation here and a more robust consideration of factors like behavioral state as a contextual factor influencing the type and severity of response. I also think that the section below seems to paint a bit of a hopeless scenario about potential generalization across scenarios. Just because there is context-dependency it doesn't mean we can't make reasonable and transparent kinds of assumptions about context scenarios - as done in Pirotta et al. (2021) ref 108. This should be specifically noted here - the paper is referenced but this is a key point.

Section 5 is pretty light on some key points - addressed above

Referee: 3

Comments to the Author(s)

General comments:

In this interesting synthesis, Keen et al. review existing Population Consequences of Disturbance models for marine mammal species to identify emerging themes in model predictions. The review presents the principles of PCoD models very well and is a timely contribution to the wider literature as PCoD models become more popular. The paper is generally well written, and the figures are excellent.

I do, however, think that from the beginning it should be clear that, while the emerging themes may also be relevant for other species, this review is focused entirely on marine mammal science and models. While the PCoD title has been primarily used for marine mammal models, an expansive suite of models exist which focus on the impacts of disturbance on non marine mammal wildlife populations. I think that highlighting this as a marine mammal centered review from the beginning ("marine mammal" is not currently mentioned in the title or abstract) could be advantageous to avoid suggesting that the review will focus on models of population impacts of disturbance at large. If the claim is made that the findings presented from marine mammals are generally transferable, the review would benefit from providing support of this claim throughout the text using either empirical or model findings from other species. It would be particularly interesting to compare results from animals with similar life history patterns but with substantial differences in other regards, e.g., large whales and elephants (Boult et al. 2019 - doi.org/10.1111/csp2.87). Though I think that the focus on acoustic disturbance in PCoD models could complicate comparisons and that it would take a considerable amount of effort to push the article towards being more inclusive of non marine mammal species. Considering this, if sticking with marine mammals, I would be a bit more tentative in making the claim that the findings for these specific marine mammal models can be used to predict risk in other species. While concepts presented may be generally useful, such as lactation being an energetically expensive and risky period for income breeders, I think that it is important to stress that without a full consideration of species-specific behavior and physiology and the environment it will be challenging to make accurate predictions of the population-level responses to disturbance.

Throughout the text both empirical results and model findings are presented. I think as a synthesis on the results of PCoD models, it needs to abundantly clear what information is coming from what source. This is well done in the reproductive strategies section but I believe that other sections could benefit from increased clarification.

With these relatively minor changes, I believe the authors can make the paper more accessible and useful to the wider wildlife modelling and conservation community.

Detailed comments:

L40-42: Have PCoD models been used in impact assessments? Would be interesting to note if so.

L81-82: Do we know that the findings of existing models are generally true? The complicated nature of disturbance responses may make it challenging to directly apply the findings from a model developed for one species to another even if life history patterns are similar.

L85, 334, & 426: Maybe these sections could just be titled "Life-history traits", "Disturbance source characteristics", etc.? I don't think the "The importance of..." part is necessary.

L116: It would be nice to have some provided examples of these lasting effects.

L117-118 & 126: Maybe it would be worth combining these two statements about some individuals in migrating populations not migrating?

L142-143: Could add a "single" disturbed area to highlight

L168-177: Many PCoD models do not explicitly consider animal movement in their simulations. It would be nice to briefly describe to what extent (and under which scales) has movement actually been considered in PCoD models and what scales of movement are important for measuring disturbance responses.

L187: Could be good to specify such as "These sensitivities to disturbance in income breeders..."

L188-189: Is this statement supposed to be related to the lactation period? As all marine mammals can store lipids for later use to some degree, is this intended to state that capital breeders may be less sensitive to foraging losses during the lactation period as lipid used for this process has already been stored? Would be good to clarify.

L200: Could add that this could ultimately impact population abundance.

L204-207: But it should be highlighted that we know very little about the specifics of how and when these decisions are made in marine mammals, particularly cetaceans.

L231-232: When known? These predictions may be very tough to get for many species/environments.

L236: Instead of basal metabolism I would say survival as it seems that this here includes costs additional to true basal metabolism, including activity, thermoregulation, feeding, etc.

L243: they require a "relatively" higher resource acquisition rate, but not necessarily absolutely unless this rate is per unit time and mass.

L256-258: Though small animals need relatively more food per unit mass they also require less food in total, for periods when food is limited smaller animals may more easily be capable of meeting their energetic needs. There are many benefits to being small which should also be discussed in this section. See:

Goldbogen, J.A., Cade, D.E., Wisniewska, D.M., Potvin, J., Segre, P.S., Savoca, M.S., Hazen, E.L., Czapanskiy, M.F., Kahane-Rapport, S.R., DeRuiter, S.L. and Gero, S., 2019. Why whales are big but not bigger: Physiological drivers and ecological limits in the age of ocean giants. *Science*, 366(6471), pp.1367-1372.

L271-273: They also have additional costs of growth.

L276: Maybe "may be exposed" rather than just "present"?

L311: Would be nice to reference Read & Hohn 1995 here.

L315: Citation needed?

L315-317: Can cite Nabe-Nielsen et al. 2018 here as the bounce back after disturbance is visible in their porpoise simulations.

L325-330: This feels a bit tagged on the end, also if this is tied to body condition estimates, I would assume that this sort of pattern would still come out of many of the cited models, but not if it is related to differences in behavior, genetics, etc.

L329-330: What would be the population level implications of this finding?

L337: Nature is a bit vague, maybe it would be good to provide an example or short description here.

L345-348: This sentence could be reworded to be more concise.

L353: Also important that the similar habitat is of a sufficient area.

L365: "consider"

L368-381: How can these technologies be paired with PCoD models?

L384-390: These two sentences have quite a few subordinate clauses in the middle of the sentence. Some rewording could make for smoother reading.

L404-405: Second "inform" isn't needed.

L407: How are disturbances modelled to impact individuals in these different models (e.g., halting energy intake)? What are the common approaches?

L416-420: This phrasing implies that received level doesn't factor at all into responses.

L441: "sensitivity to disturbance".

L451-454: What about seasonal or temporal variations in energy intake needs which are not related to reproduction but instead the environment? e.g. the seasonality in energy balance presented in:

Gallagher, C.A., Grimm, V., Kyhn, L.A., Kinze, C.C. and Nabe-Nielsen, J., 2021. Movement and seasonal energetics mediate vulnerability to disturbance in marine mammal populations. *The American Naturalist*, 197(3), pp.296-311.

L479-481: Just influences? Would be nice to be more specific here.

L514: Were these combined effects synergistic?

L530: This section could be a bit expanded a bit to be more specific about identified data gaps across the different modelling exercises. It would be interesting to see for models in which sensitivity analyses were carried out, if there were any common themes identified for types of data that these sorts of models are particularly sensitive to.

L541-542: I would say that for the vast majority of populations this information isn't known, what are the implications of that? and how do we get around this issue?

Figure 1: It is a little difficult to read the colored text (particularly the green). I would use bold to make this more legible for those of us with inferior vision.

Figure 2a: I find it a bit challenging to read this figure. I think that, as is, it doesn't communicate the main points effectively. In the Nomadic and Migratory populations it is unclear whether tracks are from a single or multiple individuals. In the original plot in Costa et al. 2016 the individual tracks were color coded so it was clear, though in the current plot it is already difficult to tell the difference between the pink and orange color at the current size. Is it necessary to have two disturbance zones? It could be nice to have a single zone shown, maybe as a fill with high transparency, and have the individual tracks colored as in the original figure to make clear the different number of individuals being disturbed.

Figure 3: It would be nice to include variability in this visual as well around the start and end of each period to encourage the consideration of peak periods if relevant - as in the Cornell birds of the world annual cycles. Also I'm not sure that having maintenance here helps anything since it is visualized as constant year round.

Author's Response to Decision Letter for (RSPB-2021-0325.R0)

See Appendix A.

RSPB-2021-0325.R1 (Revision)

Review form: Reviewer 1

Recommendation

Accept as is

Scientific importance: Is the manuscript an original and important contribution to its field?

Excellent

General interest: Is the paper of sufficient general interest?

Excellent

Quality of the paper: Is the overall quality of the paper suitable?

Excellent

Is the length of the paper justified?

Yes

Should the paper be seen by a specialist statistical reviewer?

No

Do you have any concerns about statistical analyses in this paper? If so, please specify them explicitly in your report.

No

It is a condition of publication that authors make their supporting data, code and materials available - either as supplementary material or hosted in an external repository. Please rate, if applicable, the supporting data on the following criteria.

Is it accessible?

N/A

Is it clear?

N/A

Is it adequate?

N/A

Do you have any ethical concerns with this paper?

No

Comments to the Author

I can see the authors' great effort to address my concerns. I do not have any further concerns, and would like to see this article published as it is.

Review form: Reviewer 2 (Brandon Southall)

Recommendation

Accept with minor revision (please list in comments)

Scientific importance: Is the manuscript an original and important contribution to its field?
Good

General interest: Is the paper of sufficient general interest?
Good

Quality of the paper: Is the overall quality of the paper suitable?
Excellent

Is the length of the paper justified?
Yes

Should the paper be seen by a specialist statistical reviewer?
No

Do you have any concerns about statistical analyses in this paper? If so, please specify them explicitly in your report.
No

It is a condition of publication that authors make their supporting data, code and materials available - either as supplementary material or hosted in an external repository. Please rate, if applicable, the supporting data on the following criteria.

Is it accessible?
N/A

Is it clear?
N/A

Is it adequate?
N/A

Do you have any ethical concerns with this paper?
No

Comments to the Author

Excellent revisions and additions. The convergence of comments among the reviewers was notable to me and I think all the additional details in paper, new appendices, and references are really helpful and justified. Thank you for putting so much time and attention into these comments. The paper is much improved, I absolutely support publication, and I think it will be very important in increasing understanding and acceptance of these processes in decision-making. I have one very minor additional suggestion, which is that I think you should do just a bit more both in the introduction and in the discussion to highlight the details and implications in Appendix C and (especially) D. Because this has been a criticism of the PCOD approach (transparency and understanding of limitations/assumptions), I think it should be a bit more explicitly discussed and emphasized in terms of how this paper now provides that clarity and guidance for managers. I also think it should be made clear(er) that because of all the species and contextual differences that are well discussed, this application of the process in risk assessments is not nor ever will be a simple, uniform 'formula' but that it will need to be adapted and tuned in terms of how it is applied. You do say this but just at the end of the intro and again in the discussion, I suggest you provide these messages a little more bluntly and clearly almost speaking directly to the managers. I see these suggestions as literally a few sentences in each section and really think the rest of the paper looks fantastic and ready to go.

Review form: Reviewer 3

Recommendation

Accept with minor revision (please list in comments)

Scientific importance: Is the manuscript an original and important contribution to its field?

Acceptable

General interest: Is the paper of sufficient general interest?

Acceptable

Quality of the paper: Is the overall quality of the paper suitable?

Good

Is the length of the paper justified?

Yes

Should the paper be seen by a specialist statistical reviewer?

No

Do you have any concerns about statistical analyses in this paper? If so, please specify them explicitly in your report.

No

It is a condition of publication that authors make their supporting data, code and materials available - either as supplementary material or hosted in an external repository. Please rate, if applicable, the supporting data on the following criteria.

Is it accessible?

N/A

Is it clear?

N/A

Is it adequate?

N/A

Do you have any ethical concerns with this paper?

No

Comments to the Author

Dear authors,

After reading through the response to reviewers and edited version of the manuscript, I find this revised version to be much improved, and greatly appreciate the author's responses to the various concerns. The text is much tighter and writing much more focused on the synthesis of PCoD results in this version, which makes for smoother reading. I especially appreciated the additional sections related to Applicability to other species and PCoD model assumptions and limitations. I have only a few remaining comments:

L178 - Reproductive strategy section: I am missing in this section some mention of offspring provisioning. The mother's susceptibility to disturbance due to high lactation costs may differ if offspring have high levels of provisioning as they reach weaning age.

L275: The "As a result" implies that PCoD models are considering factors like oxygen-carrying capabilities and the impacts of experience on an animal's ability to cope with disturbance, and that emerging high rates of starvation for juveniles and young females in PCoD models are due to these factors, which, to my knowledge, isn't the case.

L411 - Nature and context of disturbance section: Since dose-response curves are mentioned in the conclusion, it would be nice if they were mentioned here in this relevant section as well.

In the Appendix PCoD model assumptions and limitations section: It would also be nice to see some mention of how most PCoD models generally only focus on single species and do not include important interspecific interactions or competition. Additionally interactions and competition within species are generally not modelled but can have important implications for the prediction of disturbance effects (see Hin, V., Harwood, J. and de Roos, A.M., 2021. Density dependence can obscure nonlethal effects of disturbance on life history of medium-sized cetaceans. *PloS one*, 16(6), p.e0252677.)

Overall, I believe that with these minor changes this timely review will be suitable for publication in Proc B.

Decision letter (RSPB-2021-0325.R1)

22-Jul-2021

Dear Ms Keen

I am pleased to inform you that your Review manuscript RSPB-2021-0325.R1 entitled "Emerging themes in Population Consequences of Disturbance models" has been accepted for publication in *Proceedings B*.

The referee(s) do not recommend any further changes. Therefore, please proof-read your manuscript carefully and upload your final files for publication. Because the schedule for publication is very tight, it is a condition of publication that you submit the revised version of your manuscript within 7 days. If you do not think you will be able to meet this date please let me know immediately.

To upload your manuscript, log into <http://mc.manuscriptcentral.com/prsb> and enter your Author Centre, where you will find your manuscript title listed under "Manuscripts with Decisions." Under "Actions," click on "Create a Revision." Your manuscript number has been appended to denote a revision.

You will be unable to make your revisions on the originally submitted version of the manuscript. Instead, upload a new version through your Author Centre.

- 1) A text file of the manuscript (doc, txt, rtf or tex), including the references, tables (including captions) and figure captions. Please remove any tracked changes from the text before submission. PDF files are not an accepted format for the "Main Document".
- 2) A separate electronic file of each figure (tiff, EPS or print-quality PDF preferred). The format should be produced directly from original creation package, or original software format. Please note that PowerPoint files are not accepted.

3) Electronic supplementary material: this should be contained in a separate file from the main text and the file name should contain the author's name and journal name, e.g. `authorname_procb_ESM_figures.pdf`

All supplementary materials accompanying an accepted article will be treated as in their final form. They will be published alongside the paper on the journal website and posted on the online figshare repository. Files on figshare will be made available approximately one week before the accompanying article so that the supplementary material can be attributed a unique DOI. Please see: <https://royalsociety.org/journals/authors/author-guidelines/>

4) Data-Sharing and data citation

It is a condition of publication that data supporting your paper are made available. Data should be made available either in the electronic supplementary material or through an appropriate repository. Details of how to access data should be included in your paper. Please see <https://royalsociety.org/journals/ethics-policies/data-sharing-mining/> for more details.

<http://datadryad.org/submit?journalID=RSPB&manu=RSPB-2021-0325.R1> which will take you to your unique entry in the Dryad repository.

Once again, thank you for submitting your manuscript to Proceedings B and I look forward to receiving your final version. If you have any questions at all, please do not hesitate to get in touch.

Sincerely,
Professor Gary Carvalho
Editor, Proceedings B
<mailto:proceedingsb@royalsociety.org>

Reviewer(s)' Comments to Author:

Referee: 1

Comments to the Author(s)

I can see the authors' great effort to address my concerns. I do not have any further concerns, and would like to see this article published as it is.

Referee: 3

Comments to the Author(s)

Dear authors,

After reading through the response to reviewers and edited version of the manuscript, I find this revised version to be much improved, and greatly appreciate the author's responses to the various concerns. The text is much tighter and writing much more focused on the synthesis of PCoD results in this version, which makes for smoother reading. I especially appreciated the additional sections related to Applicability to other species and PCoD model assumptions and limitations. I have only a few remaining comments:

L178 - Reproductive strategy section: I am missing in this section some mention of offspring provisioning. The mother's susceptibility to disturbance due to high lactation costs may differ if offspring have high levels of provisioning as they reach weaning age.

L275: The “As a result” implies that PCoD models are considering factors like oxygen-carrying capabilities and the impacts of experience on an animal’s ability to cope with disturbance, and that emerging high rates of starvation for juveniles and young females in PCoD models are due to these factors, which, to my knowledge, isn’t the case.

L411 - Nature and context of disturbance section: Since dose-response curves are mentioned in the conclusion, it would be nice if they were mentioned here in this relevant section as well.

In the Appendix PCoD model assumptions and limitations section: It would also be nice to see some mention of how most PCoD models generally only focus on single species and do not include important interspecific interactions or competition. Additionally interactions and competition within species are generally not modelled but can have important implications for the prediction of disturbance effects (see Hin, V., Harwood, J. and de Roos, A.M., 2021. Density dependence can obscure nonlethal effects of disturbance on life history of medium-sized cetaceans. *PloS one*, 16(6), p.e0252677.)

Overall, I believe that with these minor changes this timely review will be suitable for publication in Proc B.

Referee: 2

Comments to the Author(s)

Excellent revisions and additions. The convergence of comments among the reviewers was notable to me and I think all the additional details in paper, new appendices, and references are really helpful and justified. Thank you for putting so much time and attention into these comments. The paper is much improved, I absolutely support publication, and I think it will be very important in increasing understanding and acceptance of these processes in decision-making. I have one very minor additional suggestion, which is that I think you should do just a bit more both in the introduction and in the discussion to highlight the details and implications in Appendix C and (especially) D. Because this has been a criticism of the PCOD approach (transparency and understanding of limitations/assumptions), I think it should be a bit more explicitly discussed and emphasized in terms of how this paper now provides that clarity and guidance for managers. I also think it should be made clear(er) that because of all the species and contextual differences that are well discussed, this application of the process in risk assessments is not nor ever will be a simple, uniform 'formula' but that it will need to be adapted and tuned in terms of how it is applied. You do say this but just at the end of the intro and again in the discussion, I suggest you provide these messages a little more bluntly and clearly almost speaking directly to the managers. I see these suggestions as literally a few sentences in each section and really think the rest of the paper looks fantastic and ready to go.

Author's Response to Decision Letter for (RSPB-2021-0325.R1)

See Appendix B.

Decision letter (RSPB-2021-0325.R2)

29-Jul-2021

Dear Ms Keen

I am pleased to inform you that your manuscript entitled "Emerging themes in Population Consequences of Disturbance models" has been accepted for publication in Proceedings B.

Your article has been estimated as being 15 pages long. Our Production Office will be able to confirm the exact length at proof stage.

Data Accessibility section

Open Access

Paper charges

Sincerely,

Proceedings B

Appendix A

UNIVERSITY OF CALIFORNIA, SANTA CRUZ

BERKELEY • DAVIS • IRVINE • LOS ANGELES • MERCED • RIVERSIDE • SAN DIEGO • SAN FRANCISCO

SANTA BARBARA • SANTA CRUZ

DEPARTMENT OF ECOLOGY & EVOLUTIONARY BIOLOGY
DIVISION OF PHYSICAL & BIOLOGICAL SCIENCES
COASTAL BIOLOGY BUILDING
130 MCALLISTER WAY
SANTA CRUZ, CALIFORNIA 95060

June 13, 2021

Dear Editorial Board,

We are pleased to submit our revised manuscript “**Emerging themes in Population Consequences of Disturbance models**” (RSPB-2021-0325) for consideration as an evidence synthesis in *Proceedings of the Royal Society B*.

We are grateful for the thoughtful reviews provided by the Associate Editor and referees. We have incorporated the requested revisions and responded to each comment in the postscript of this letter, in blue. We have also included a copy of the manuscript with revisions as tracked changes.

I can be reached by email (kelly.a.keen@gmail.com) or by phone (+1 215 287 8530). My co-authors' email addresses are listed below:

Roxanne Beltran: roxanne@ucsc.edu

Enrico Pirotta: pirotta.enrico@gmail.com

Daniel Costa: costa@ucsc.edu

Yours sincerely,

Kelly Ann Keen

PhD Student

Ecology and Evolutionary Biology

University of California, Santa Cruz

Associate Editor Comments

Thank you for submitting your interesting manuscript for consideration as a PRSB Evidence Synthesis article. I would like to start out by apologising the time it has taken to secure appropriately high quality and representative referee reports, but as you will see below, despite the delay, I hope that you find the level of detail and constructive tone of significant help in taking your work forward.

Your work has now been reviewed by 3 experts in the field, and I have read the manuscript myself. While there is a consensus that the topic is pertinent for an evidence synthesis article, and timely in that it is likely to command broad interest, you will see below some detailed and constructive suggestions of how to improve your MS further. You will see that a range of suggestions indicating collectively that your manuscript and topic, is in my opinion, worthy of additional investment and consideration, and accordingly invite a revision.

As such, I encourage you strongly to consider the comments below, and submit a revised manuscript at your earliest convenience. You will see that there are a variety of concerns which I echo, and highlight here, though full details are provided in the respective referee reports. A fundamental concern cross referees, which I endorse, is the extent to which your consideration can be applied across non-marine mammal taxa. I appreciate that this will require additional text, though indicating the extent to which such an approach is transferable, with brief illustrations of some examples, or indeed, how and why generic value is constrained, would be of profound interest to the wide readership of PRSB. To what extent has this kind of model been applied to other taxa? Some sharpening of the justification for underlying assumptions of the approach is also required, as well as the extent to which these types of models may, or may not, have a common overall structure or mathematical form. The latter may be most effectively summarised in a Figure, as proposed below, alongside other existing highly informative schematics. Enhanced accessibility to our readership of your approach and applications, would be enhanced by such additional information, without necessarily increasing the length significantly.

It is also important to provide a more robust and justified rationale for the choice of factors that need to be considered, especially in the section of environmental conditions, with some specific suggested factors given below. Indeed, as one of the referees highlights, is a major suggestion, a more substantive critique on the underlying prior assumptions which have been well or poorly parameterised by data, including previous limitations of PCOD assessments would be of significant benefit. It is of course vitally important in evidence synthesis articles, wherever possible, to emphasise practical utility and translation of empirical data/evidence, into practical policy and in this case, facilitating managers in the context of risk assessment frameworks. One final general point, initial to the numerous helpful suggestions, is the methodology employed in accessing and synthesising the evidence base presented. It is especially vital within evidence synthesis manuscripts to provide a robust, transparent and representative framework with explicit clarity on the source and choice of information presented. While this is relatively clear in the section on reproductive strategies, it is perceived to be less so elsewhere.

Much of the content of the constructive referee reports is self-evident, and I hope should you decide to revise your manuscript, will provide a useful guide of how to proceed. I would like to

additionally draw your attention specifically to our requirements for publication of Evidence Synthesis articles. Notwithstanding, in your response to referees, I would be grateful if you would include a brief account relating to Editorial comments, on how the manuscript has been modified in relation to my brief suggestions. In particular, as you will have seen from the guidelines available for our Evidence Synthesis articles (<https://royalsocietypublishing.org/rspb/evidence-synthesis>), it is vital that the reader is able to assess the validity, robustness and objectivity of the evidence base presented. Importantly also, when putting the final touches to the article, please ensure wherever possible, that where relevant, you have addressed some of the questions below, that characterises the Evidence Synthesis article type, though I fully recognise, that many questions will only partially apply to your manuscript:

1. Is the key policy-related question(s) articulated clearly?
2. Is there a clear justification in support of policy relevance?
3. Is the likely target audience identified clearly?
4. Does the search for literature utilise a comprehensive range of sources?
5. Does the synthesis article apply clearly documented inclusion criteria to all potentially relevant studies found during the search?
6. Is a clear methodology described to avoid bias?
7. Is your study objectively weighted according to methodological quality of cited literature?
8. Are knowledge gaps and priorities clearly identified?
9. Are outcomes/recommendations tangible in terms of likely impact?
10. Are all necessary supporting information available and accessible??

Including a brief indication of how you have addressed the specific criteria above in your response letter would be most helpful. I appreciate that the volume of revision is extensive, and may go beyond what you had originally anticipated. Notwithstanding, I would hope you will find the constructive and detailed suggestions helpful in formulating a more robust and representative evidence synthesis article for resubmission. As indicated below, as in all peer review processes, the invitation to revise, is of course no guarantee of eventual publication, but I will do my best to exercise consistency in the remaining peer review process, by approaching the original referees, at a minimum, though of course I am not in a position to confirm their availability.

Thank you in advance for bringing this information together, and we look forward to receiving the revised manuscript in due course.

We appreciate the editor's kind words about the manuscript. Below, we provide a brief summary of how we addressed the major comments highlighted in the editorial comments:

- **Applicability across taxa:** We added the section "Applicability to other species" to the supplementary material to describe how the PCoD framework and the emerging themes from existing models can be applied to other species. While we would have preferred to put this in the manuscript, we included it in the supplementary material to meet the journal's page limit for evidence-based syntheses.

- PCoD model assumptions and limitations: We added the section “PCoD model assumptions and limitations” to the supplementary material, which describes the common assumptions and limitations to PCoD models. While we would have preferred to put this in the manuscript, we included it in the supplementary material to meet the journal’s page limit for evidence-based syntheses.
- Common overall structure/mathematical form: The PCoD framework is a conceptual model that can be formalized in many different ways, as discussed in the “Introduction” (e.g., via matrix models, bioenergetic modelling, stochastic dynamic programming), and the functional form changes across applications depending on the data available and the questions being addressed. As we have limited space in the main text, we added a visual representation of the PCoD framework to the supplementary material (see Figure S1) to describe how a disturbance source can lead to changes in behaviour and physiology that can affect health and vital rates and, ultimately, population dynamics. In addition, we have included a sentence in the third paragraph of the “Introduction” directing the reader to a review by Pirotta et al. (2018) that describes how empirical data and alternative methods have been used to parameterize PCoD models.
- Rationale for factors considered, particularly in the “Environmental conditions” section: In the “Introduction,” we added a sentence regarding the criteria used to select PCoD models for the synthesis. We also highlight in the “Introduction” and then in the “PCoD model assumptions and limitations” section in the supplementary material that most implementations of the PCoD framework have focused on the behavioural-bioenergetic pathway (i.e., mediated by changes in energy acquisition or expenditure, which we focus on in this synthesis). However, stressors may operate along other pathways that may not act via changes in energy budget, as one of the reviewer’s points out, and may elicit physiological responses that can affect individual health and vital rates. As such, the themes that have emerged from the models included in the synthesis are the result of this focus on the behavioural-bioenergetic pathway.
- Practical utility of the emerging themes: In each section, we describe how the information discussed can be applied to assess risk and often suggest general principles or tools (e.g., reproductive cycle plots, Important Marine Mammal Areas or IMMAs) that can assist with such analyses. We also added text to the last paragraph of the manuscript to describe how the findings from this synthesis can be used and how future PCoD models can add to our findings.
- Methodology: We added text to the “Introduction” to describe how PCoD models were selected for the synthesis. See answers to questions #4 and #5 below.

Below, we also include a brief summary of how we have addressed many criteria for an evidence synthesis. The line numbers refer to those in the manuscript with tracked changes.

1. The key policy-related question is articulated in the “Abstract” (lines 13-18, 21-23, and 25-27) and “Introduction” (lines 90-170). We describe how the PCoD framework was developed to conceptually link how disturbance can lead to changes in population dynamics, and the real-world application of the framework

has led to a suite of quantitative models that can inform risk assessments. However, the findings from these disparate models have yet to be synthesized in the single review that can be used as a reference by wildlife managers, practitioners, and industry when assessing risk. Such information can help identify and prioritize the populations most vulnerable to disturbance in risk assessments and guide the planning of activities that avoid or mitigate population-level effects. Our synthesis fills that gap.

2. PCoD models have been used to forecast the possible consequences of a range of disturbance scenarios and identify the disturbance level likely to result in a population impact. By synthesizing these model findings, important contextual factors can be identified that can help wildlife managers and practitioners identify and prioritize the populations most vulnerable to disturbance in risk assessments and guide industry in planning activities that avoid or mitigate population-level effects (e.g., lines 98-137). The information in this synthesis can be used as a reference by wildlife managers, practitioners, and industry when assessing risk. In addition, this information can be used to inform decision frameworks, such as the framework developed by Wilson et al. (2019) which provides a cost-effective, hierarchical approach to assessing the impact of disturbance at the population level (e.g., lines 135-137).
3. The target audience, which includes wildlife managers, practitioners, and industry, is identified clearly throughout the synthesis (e.g., lines 25-27, 135-137).
4. To identify PCoD models for this synthesis, we searched Google Scholar to gather studies from 2005 (when the PCoD framework was first published) through March 2021 using specific search terms. A review paper by Pirotta et al. (2018) and the papers that met the synthesis criteria (see the answer to question #5, below) were also used to manually extract additional references that were not identified using the search terms. Our process for selecting models for the synthesis is described in the “Introduction” (lines 148-155).
5. Our inclusion criteria are included in the last paragraph of the “Introduction” (lines 152-154) and in Table S1 in the supplementary material. To be included in the synthesis, models had to assess the population consequences of disturbance for marine mammals and quantify how non-lethal anthropogenic disturbance can affect vital rates via the behavioural-bioenergetic pathway. The models that met these criteria are listed in Table S1 of the supplementary material.
6. Our search methodology and inclusion criteria in #4 and #5, above, were employed to avoid bias in selecting the models included in the synthesis.
7. Not applicable.
8. We clearly identify knowledge gaps and future priorities in the “Data gaps and future priorities” section (beginning on line 913).

9. We provide recommendations throughout the paper for how the target audience can apply the emerging themes to identify priority/vulnerable species and to assess risk (e.g., reproductive cycle plots (i.e., lines 417-419, Figure 3), spatial management tools (lines 630-680)). In the “Data gaps and future priorities” section, we also provide recommendations to improve real-world model application, emphasizing the need for partnerships between modellers and wildlife managers to identify ways to increase model accessibility and for transparency about prior assumptions that may affect model outputs and permit/policy decisions (lines 917-920). In addition, we recommend developing models for representative populations or species exposed to common disturbance scenarios to investigate broad patterns in population responses to disturbance. By identifying population characteristics and other contextual factors that could lead to population-level effects, the findings from these models could be used to guide decision making and develop mitigation strategies that target populations most at risk or sensitive to a proposed activity (lines 992-997).
10. All supporting information is available (i.e., published) and accessible. In the supplementary material, we include a list of the PCoD models used to identify the emerging themes in this synthesis (see Table S1).

Referee 1 Comments

This is a well written review paper that describe and discuss an important type of models. In this synthesis the authors reviewed common themes about Population Consequences of Disturbance (PCoD) models. They highlighted essential factors that need to be considered in this kind of models and identified data gaps and suggested future directions. To be honest, I am not an expert in this kind of models and thus would not be able to provide in-depth comments. I do get very excited about this kind of models by reading this manuscript, and believe this article will be very useful for many people like myself to learn more about this kind of models. Therefore, from my point of view I would love to see this article published in a high-level journal such as Proceedings B. I nonetheless have a few very minor comments for the authors to consider.

We appreciate the reviewer’s kind words.

Line 83-84. “... although we focus on marine mammals, these general principles could guide risk assessments for other wildlife species.” I understand that the authors have focused on marine mammals (which is totally fine). I am just curious if this kind of models have been applied to other taxa (e.g., terrestrial large mammals). I think it will be great if the authors could provide just a few examples (citations) for applications in taxa other than marine mammals. If such examples do not exist, what would be the potential challenges to apply this kind of models to other taxa? It will be great if the authors could briefly discuss this kind of issues (and my apologies if the authors have already done so but I overlooked it).

We added the section “Applicability to other species” to the supplementary material to describe how the PCoD framework and the emerging themes from existing models can be applied to other species. This information was included in the supplementary material to

meet the journal's page limit for evidence-based syntheses. Below, we have included an excerpt from the new section to address the reviewer's comment:

The Population Consequences of Disturbance (PCoD) framework is a conceptual model that can be used to assess the population consequences of non-lethal changes in behaviour and physiology. Although originally developed to assess the risks to marine mammal populations from anthropogenic noise (particularly naval sonar), it has since been used to explore the effects of other disturbance sources (e.g., wildlife tourism, renewable energy developments, vessels) and can, in principle, be applied to other vertebrate species. The literature on the effects of anthropogenic disturbance on the behaviour of non-marine mammal taxa is extensive [147-149]. In addition, many studies have linked disturbance-induced changes in behaviour to reproductive success and survival [150-152] and quantified the long-term effects on population dynamics [153-156]. As such, these studies could be reformulated following the PCoD framework.

It seems that this kind of models can have flexible structures and can incorporate multiple types of information. Do these models have a common overall structure or even a common mathematical form? If they could take a common overall structure, it would be great if the authors could provide a figure to illustrate that. I appreciate the nice figures the authors provided, but it seems there lacks something that can guide the readers to start developing such a model.

The PCoD framework is a conceptual model that can be formalized in many different ways, as discussed in the "Introduction" (e.g., via matrix models, bioenergetic modelling, stochastic dynamic programming), and the functional form changes across applications depending on the data available and the questions being answered. As we have limited space in the main text, we added a visual representation of the PCoD framework to the supplemental (see Figure S1) to describe how a disturbance source can lead to changes in behaviour and physiology that can affect health and vital rates and, ultimately, population dynamics. In addition, we have included a sentence in the third paragraph of the "Introduction" directing the reader to a review by Pirotta et al. (2018) that describes how empirical data and alternative methods have been used to parameterize PCoD models.

I again believe that the manuscript is well organized, and Figure 1 in particular summarize and help the readers to understand the main points very well. I just wonder if there are other factors that need to be considered in the "Environmental Conditions" section. In particular, I would think that human disturbances such as hunting/poaching is still a major threat to marine mammals. Other things I can think about include pollution (e.g., plastics) and collision between animals and ocean vessels. Can these things be considered in this kind of models? If so, shall these be discussed in addition to prey abundance and climate change? I think discussing these things are important as terrestrial animals may face similar issues (e.g., poaching, traffic).

We appreciate the reviewer's kind words about the manuscript and Figure 1. We also appreciate the reviewer's mention of additional threats in the environment that marine mammals face, including hunting/poaching, collisions with vessels, and pollution (e.g., plastic). The PCoD framework was developed to help quantify the effects of *non-lethal* changes in behaviour and physiology, such as shifts in habitat use or increased levels of stress, on a population. Because hunting/poaching leads directly to death, the potential

effects on a population are relatively easier to quantify. Since collisions with vessels and chemical pollution could lead to non-lethal changes, they can be incorporated into a model as sources of disturbance. Most applications of the PCoD framework have focused on the behavioural-bioenergetic pathway (i.e., mediated by changes in energy acquisition or expenditure, which we focus on in this synthesis), but there are other stressors, as the reviewer points out, that may not act via changes in energy budget, and may elicit physiological responses that can affect individual health and vital rates. In the “Data gaps and future priorities” section, we highlight that more research is needed to quantify disturbance-related changes in physiology that compromise individual health, as well as the need for baseline and long-term measurements of health and how they relate to variation in vital rates. We also highlight this as an important assumption/limitation in the “PCoD model assumptions and limitations” section of the supplementary material.

Referee 2 Comments

Because the fundamental approach uses bioenergetic models as the basis of PCOD, I would like to see some more explicit description of whether and how such a perspective has been applied in terms of describing and predicting population demographics in other taxa. I know this has been done in non-marine mammals and some examples could help substantiate the use of bioenergetics to populations within the context of disturbance here.

We added the section “Applicability to other species” to the supplementary material to describe how the PCoD framework can be applied to other species. This information was included in the supplementary material to meet the journal’s page limit for evidence-based syntheses. Below, we have included an excerpt from the new section to address the reviewer’s comment:

The Population Consequences of Disturbance (PCoD) framework is a conceptual model that can be used to assess the population consequences of non-lethal changes in behaviour and physiology. Although originally developed to assess the risks to marine mammal populations from anthropogenic noise (particularly naval sonar), it has since been used to explore the effects of other disturbance sources (e.g., wildlife tourism, renewable energy developments, vessels) and can, in principle, be applied to other vertebrate species. The literature on the effects of anthropogenic disturbance on the behaviour of non-marine mammal taxa is extensive [147-149]. In addition, many studies have linked disturbance-induced changes in behaviour to reproductive success and survival [150-152] and quantified the long-term effects on population dynamics [153-156]. As such, these studies could be reformulated following the PCoD framework.

My main suggestion relates to the need to specifically consider and honestly discuss the underlying prior assumptions, which have been well or poorly parameterized by data, and what the limitations of previous PCOD assessments are. One of the primary criticisms of and limitations to how much impact these modeling approaches have actually had in management decisions is that they are not sufficiently transparent in their assumptions and which are strongly or weakly supported and that the underlying mathematical processes are not clear and replicable for non-specialists (like decision makers). Is this a fair criticism? I kind of think it is to be honest but the authors may not. Either way, I'd strongly suggest this be directly and explicitly addressed

in the intro and then especially picked up in data gaps and future priorities. The parameterization discussion is there and I do like the discussion of limitations of PBR (even though it is quite US centric). But the reader is left wanting at the end in my opinion about how best to push these fantastic conclusions from the emerging themes (I love Fig 1 - I'm going to put it on my wall) really and practically into play. Specifically, how will advancing PCoD and PCoMS help managers - last sentence? How specifically do you think these tools should be applied - risk assessment frameworks, conceptual models, within biological opinions as a reference? We can't expect a resource analysts at NMFS or BOEM or JNCC to be able to run or even fully understand the details of these complex models. This is an impediment to their direct application. While I think this paper goes a long, long way into extracting the key points (again Fig 1) but I just encourage you to specifically suggest how best people in those scenarios that aren't versed in the details of the stats should really try and actually apply something like the recent PDoD results in a practical scenario.

We appreciate the reviewer's kind words about the manuscript and Figure 1.

Regarding model assumptions and limitations, we added the section "PCoD model assumptions and limitations" to the supplementary material. This information was included in the supplementary material to meet the journal's page limit for evidence-based syntheses. Some of the common assumptions/limitations discussed include:

- PCoD models include assumptions about model parameters and underlying mechanisms that may influence model outputs, but the uncertainty associated with such assumptions can be addressed and quantified.
- Most PCoD models have focused on the behavioral-bioenergetic pathway, but some stressors may operate along other pathways, including those that trigger physiological responses.
- Most PCoD models have also only considered sources of disturbance in isolation, necessitating the need to extend the framework to consider the population consequences of multiple stressors via the PCoMS framework.
- Most PCoD models have not explicitly considered individual heterogeneity, which can affect the overall sensitivity of a population to disturbance.
- The scale mismatch between the data and the management or policy issue being addressed (often fine) and the scale of the model (often coarse), which can affect the scale at which model outputs can be applied to inform conservation and management decisions.

Regarding limitations in real-world model application, we included recommendations in the "Data gaps and future priorities" section emphasizing the need for partnerships between modellers and wildlife managers to identify ways to increase model accessibility and the outputs necessary to make decisions. We also recommend that model assumptions should be made transparent to end users, in addition to any impact they may have on model outputs and permit/policy decisions. We believe this will help non-specialists better understand the uncertainty associated with some model outputs. In the interim, we believe the emerging themes from PCoD models discussed in this synthesis and the recommendations provided throughout the paper and summarized in Figure 1 can

be used by non-specialists to assess risk and can be readily incorporated into risk assessments and biological opinions. Further, in the “Data gaps and future priorities” section, we recommend the development of models for representative populations or species exposed to common disturbance scenarios to investigate broad patterns in population responses to disturbance. By identifying population characteristics and other contextual factors that could lead to population-level effects, the findings and common themes from these models could be used to guide decision making and develop mitigation strategies that target populations most at risk or sensitive to a proposed activity.

Abstract- lines 13-15. The second part of this sentence is really important and may not be inherently familiar in terms of what you mean to readers of PRSB. I suggest you make this a separate and simple sentence and emphasize the important role that such quantitative models can play in better parameterizing probabilistic risk assessments for assessing the longer-term severity of disturbance.

We appreciate the reviewer’s suggestion. We edited the first couple of sentences in the “Abstract” as follows: “Assessing the non-lethal effects of disturbance from human activities is necessary for wildlife conservation and management. However, linking short-term responses to long-term impacts on individuals and populations is a significant hurdle for evaluating the risks of a proposed activity. The Population Consequences of Disturbance (PCoD) framework conceptually describes how disturbance can lead to changes in population dynamics, and its real-world application has led to a suite of quantitative models that can inform risk assessments.”

Line 19. I would include 'assumptions' and 'limitations' around the word 'findings' (see general comments on the need for this in my view

We included the words “assumptions” and “limitations” in the “Abstract.” The sentence reads: “We also discuss model assumptions and limitations, identify data gaps, and suggest future research directions to enable PCoD models to better inform risk assessments and conservation and management decisions.”

Abstract. Somewhere in the second half of the abstract I think the term 'transparent' or 'evident to non-specialists' or non-statisticians should appear. One of the main limitations I have seen in practical applications is that the priors and assumptions are not readily evident and transparent (also addressed above).

We added the phrase “self-evident to non-specialists or non-statisticians” to the second-to-last paragraph of the “Introduction.” The sentence reads: “Finally, in the supplementary material, we discuss how the PCoD framework and the emerging themes in this synthesis can be broadly applied to guide risk assessments for other species (see *Applicability to other species*), as well as underlying model assumptions and limitations that may not be self-evident to non-specialists or non-statisticians (see PCoD model assumptions and limitations).” As the “Abstract” has a word limit, we are unable to include it there.

Abstract. There is no indication of taxa emphasis within the abstract - the term marine mammals is not used but should be woven in somewhere and briefly why this is the case.

We added “marine mammals” to the “Abstract.” The sentence reads: “Here, we review PCoD models that forecast the possible consequences of a range of disturbance scenarios for marine mammals.” In the “Introduction,” we also included that the PCoD framework was developed for use with marine mammals. The sentence reads: “While this framework was developed for use with marine mammals, it is generally applicable across most vertebrates.” In addition, we included text in the “Applicability to other species” section in the supplementary material that describes how the PCoD framework was originally developed for marine mammals to assess the risks associated with anthropogenic noise. The sentence reads: “Although originally developed to assess the risks to marine mammal populations from anthropogenic noise (particularly naval sonar), it has since been used to explore the effects of other disturbance sources (e.g., wildlife tourism, renewable energy developments, vessels) and can, in principle, be applied to other vertebrate species.”

Intro - first paragraph. Would like to see a more recent reference than NRC 2005 and many more updated broad reviews. Additionally, later in the paragraph, a good recent reference spanning national and international policy implications of this issue is: Chou, E., et al. (2021). International policy, recommendations, actions and mitigation efforts of anthropogenic underwater noise. *Ocean & Coastal Management*, 202, 105427.

We edited this sentence to capture some of the behavioural and physiological responses exhibited by vertebrates to human activities. As such, we removed NRC (2005) and NAS (2017), and we included the following:

- Pirotta, E, Booth, CG, Costa, DP, Fleishman, E, Kraus, SD, Lusseau, D, Moretti, D, New, LF, Schick, RS, Schwarz, LK, et al. 2018 Understanding the population consequences of disturbance. *Ecol. Evol.* 8, 9934-9946.
- Duarte, CM, Chapuis, L, Collin, SP, Costa, DP, Devassy, RP, Eguiluz, VM, Erbe, C, Gordon, TA, Halpern, BS & Harding, HR. 2021 The soundscape of the Anthropocene ocean. *Science* 371.
- Wilson, MW, Ridlon, AD, Gaynor, KM, Gaines, SD, Stier, AC & Halpern, BS. 2020 Ecological impacts of human-induced animal behaviour change. *Ecol. Lett.* 23, 1522-1536.

We also added Chou et al. (2021) as a reference later in the paragraph where national and international legislation is discussed.

Line 53. Very good to mention whale watching here. In case any other reviews balk at this, I believe it is a very apt inclusion. Could arguably add elevated background noise in high traffic areas as well.

We appreciate the reviewer’s feedback.

Figure 1. Fantastic. This is the most important and impactful set of messages in the paper.

We appreciate the reviewer's kind words about Figure 1.

Line. 94 - Section on movement ecology. Lot of good and specific references here. One important one that reviewed this topic across a number of taxa and made a very clear point earlier along these lines is Forney et al. 2017. This could be used instead of several of the many references here (there are quite a lot of refs in the paper overall and if anything this section is more detailed than I think needs to be), but it definitely should appear here somewhere as it was one of the first papers to make this overall point about movement and susceptibility to disturbance. Forney, K. A., et al. (2017). Nowhere to go: noise impact assessments for marine mammal populations with high site fidelity. *Endangered Species Research*, 32, 391-413.

We added Forney et al. (2017) as a reference at the end of the first sentence in the "Movement ecology" section. We also made additional edits to this section to reduce the amount of detail.

p. 178 Section on Reproductive strategies. This section is better focused and tighter in the points made than the one above on movement. I suggest mirroring the level of detail and focus to that in this section

We appreciate the reviewer's feedback. We believe much of the detail in the "Movement ecology" section is important to consider when assessing risk but have made edits to reduce text/examples.

Lines 408-413. Nature and context. In contrast, this section is really light on something that is fundamentally and even overarchingly important at least in terms of acute responses. There should be some discussion of novelty and habituation here and a more robust consideration of factors like behavioral state as a contextual factor influencing the type and severity of response. I also think that the section below seems to paint a bit of a hopeless scenario about potential generalization across scenarios. Just because there is context-dependency it doesn't mean we can't make reasonable and transparent kinds of assumptions about context scenarios - as done in Pirota et al. (2021) ref 108. This should be specifically noted here - the paper is referenced but this is a key point.

We agree with the reviewer – novelty and habituation are important factors that influence the type and severity of response and should be considered in both risk assessments and PCoD models. We have included two sentences highlighting the importance of these factors, and emphasize that, while important, they have yet to be incorporated into PCoD models. The sentences read: "An individual's experience can also influence the severity of response, although changes in responsiveness have yet to be incorporated into PCoD models. For example, a novel disturbance event may cause an overt reaction, while prior experience may lead to habituation or sensitization [94, 104]." These factors are also included as data gaps/future priorities for PCoD models in the "Data gaps and future priorities" section (see lines 924-925 in manuscript with tracked changes) and discussed

in the “PCoD model assumptions and limitations” section (see fourth paragraph) in the supplementary material.

Regarding context-dependency, some generalizations can be made as the evidence base is expanding, but we are far from being able to model these processes explicitly for most species. We have included text that some generalizations can be made (as demonstrated in this synthesis), but that an improved understanding of the underlying processes for how and why individuals respond to disturbance will allow for more accurate predictions. The sentences read: “Disturbance sources may have radically different effects depending on both intrinsic and extrinsic factors, making it difficult to compare across scenarios. Nevertheless, generalizations can be made (as demonstrated in this synthesis) regarding which factors may have the greatest effects, and an improved understanding of the underlying processes for how and why individuals respond to disturbance may allow for more accurate predictions.”

Section 5 is pretty light on some key points - addressed above

Please see our response to the reviewer’s general comments provided above.

Referee 3 Comments

In this interesting synthesis, Keen et al. review existing Population Consequences of Disturbance models for marine mammal species to identify emerging themes in model predictions. The review presents the principles of PCoD models very well and is a timely contribution to the wider literature as PCoD models become more popular. The paper is generally well written, and the figures are excellent.

We appreciate the reviewer’s kind words about the manuscript text and figures.

I do, however, think that from the beginning it should be clear that, while the emerging themes may also be relevant for other species, this review is focused entirely on marine mammal science and models. While the PCoD title has been primarily used for marine mammal models, an expansive suite of models exist which focus on the impacts of disturbance on non marine mammal wildlife populations. I think that highlighting this as a marine mammal centered review from the beginning (“marine mammal” is not currently mentioned in the title or abstract) could be advantageous to avoid suggesting that the review will focus on models of population impacts of disturbance at large. If the claim is made that the findings presented from marine mammals are generally transferable, the review would benefit from providing support of this claim throughout the text using either empirical or model findings from other species. It would be particularly interesting to compare results from animals with similar life history patterns but with substantial differences in other regards, e.g., large whales and elephants (Boult et al. 2019 - doi.org/10.1111/csp.2.87). Though I think that the focus on acoustic disturbance in PCoD models could complicate comparisons and that it would take a considerable amount of effort to push the article towards being more inclusive of non marine mammal species. Considering this, if sticking with marine mammals, I would be a bit more tentative in making the claim that the findings for these specific marine mammal models can be used to predict risk in other species.

While concepts presented may be generally useful, such as lactation being an energetically expensive and risky period for income breeders, I think that it is important to stress that without a full consideration of species-specific behavior and physiology and the environment it will be challenging to make accurate predictions of the population-level responses to disturbance.

We appreciate the reviewer's feedback. We added "marine mammals" to the "Abstract" to emphasize the paper's focus. The sentence reads: "Here, we review PCoD models that forecast the possible consequences of a range of disturbance scenarios for marine mammals."

We also added the section "Applicability to other species" to the supplementary material to describe how the PCoD framework and the emerging themes from existing models can be applied to other species. This information was included in the supplementary material to meet the journal's page limit for evidence-based syntheses. Below, we have included text from the new section to address the reviewer's comment:

The Population Consequences of Disturbance (PCoD) framework is a conceptual model that can be used to assess the population consequences of non-lethal changes in behaviour and physiology. Although originally developed to assess the risks to marine mammal populations from anthropogenic noise (particularly naval sonar), it has since been used to explore the effects of other disturbance sources (e.g., wildlife tourism, renewable energy developments, vessels) and can, in principle, be applied to other vertebrate species. The literature on the effects of anthropogenic disturbance on the behaviour of non-marine mammal taxa is extensive [147-149]. In addition, many studies have linked disturbance-induced changes in behaviour to reproductive success and survival [150-152] and quantified the long-term effects on population dynamics [153-156]. As such, these studies could be reformulated following the PCoD framework.

Many of the emerging themes reviewed in this synthesis are not unique to marine mammals and can be used to guide risk assessments for other vertebrate species. For example, we describe movement patterns that are prevalent across widely disparate taxa, regardless of body size, mode of movement, or environment [157], and can be used to determine the degree of exposure to a disturbance-inducing activity. If exposure is likely, sensitivity can be assessed by understanding how parents acquire and allocate resources to offspring. The concepts of capital and income breeding have been used to describe reproduction in other taxa [158-161], and where a population falls on this continuum can be used to determine whether a proposed activity overlaps with sensitive reproductive stages. Body size can also help assess sensitivity to disturbance, particularly among vertebrates where an individual's mass-specific metabolic rate decreases with increasing body size [162]. Thus, a larger body size may provide a temporary buffer during disturbance events that result in reduced or lost foraging opportunities. Pace of life is also a concept that has been broadly applied across vertebrates [163], and can be used to assess how disturbance can affect population growth via changes in vital rates.

The influence of disturbance source characteristics and environmental conditions emerging from this synthesis is also broadly applicable across vertebrate species, although the specific tools (e.g., the International Union for Conservation of Nature's

(IUCN) Important Marine Mammal Area's) or environmental processes (e.g., El Niño Southern Oscillation) discussed may be specific to marine mammals or to marine or coastal environments. Nevertheless, similar spatial tools exist in terrestrial and freshwater habitats (e.g., the United Nation's Environment Programme's and IUCN's World Database on Protected Areas, BirdLife International's Important Bird and Biodiversity Areas) and can be used to assess the overlap between a proposed activity and important habitats for critical life-history stages. Additionally, environmental conditions, such as interannual variability [164] and drought events [165], can influence female body condition and impact reproductive success in other vertebrates. Considering such conditions, when known, can guide activity planning that reduces overlap with sensitive periods. Ultimately, limiting or avoiding exposure to repeated or continuous disturbance in biologically important habitats and during periods of reproductive sensitivity and low prey availability may reduce the potential for population-level effects.

Throughout the text both empirical results and model findings are presented. I think as a synthesis on the results of PCoD models, it needs to abundantly clear what information is coming from what source. This is well done in the reproductive strategies section but I believe that other sections could benefit from increased clarification.

To differentiate between empirical results and PCoD model findings, we included words like simulated and modelled to sentences where PCoD model findings are discussed.

With these relatively minor changes, I believe the authors can make the paper more accessible and useful to the wider wildlife modelling and conservation community.

We appreciate the reviewer's feedback and kind words.

L40-42: Have PCoD models been used in impact assessments? Would be interesting to note if so.

In the "Data gaps and future priorities" section, we describe how, in the absence of sufficient empirical data, an interim approach to PCoD (iPCoD) has been used in risk assessments, and provide references to two examples:

- Booth, CG, Harwood J, Plunkett R, Mendes S, and Walker R. 2017. Using the Interim PCoD framework to assess the potential impacts of offshore wind developments in Eastern English Waters on harbour porpoises in the North Sea. Natural England Joint Report, Number 024 York.
- Smith, H., C. Carter, and F. Manson. 2019. Cumulative impact assessment of Scottish east coast offshore windfarm construction on key species of marine mammals using iPCoD. Scottish Natural Heritage Research Report No. 1081.

We also included King et al. (2015) as the main reference for the iPCoD approach. The sentence reads: "In the absence of sufficient empirical data, an interim PCoD approach has been used in risk assessments and parameterized via expert elicitation to quantify the relationship between changes in behaviour and physiology to fitness [19, 132, 133]."

L81-82: Do we know that the findings of existing models are generally true? The complicated nature of disturbance responses may make it challenging to directly apply the findings from a model developed for one species to another even if life history patterns are similar.

Regarding whether the findings of existing models are generally true, the emergent properties of a model can be compared with empirically derived data to ensure the model accurately represents the natural dynamics of the population/species. Sensitivity analyses may also be performed to determine the robustness of the conclusions, and uncertainty in the estimated population consequences can be reported as a distribution of potential outcomes.

Regarding applicability to other marine mammal populations/species, we agree that these species- and context-specific models are limited in their applicability across taxa and disturbance scenarios when considered in isolation. However, when considered holistically, these models can provide valuable insight into which contextual factors influence a population's degree of exposure and sensitivity to disturbance. As such, we emphasize that PCoD model results can be used for guidance (as demonstrated in this synthesis), especially when the various sources of uncertainty are appropriately explored and propagated (see the "Data gaps and future priorities" section and the "PCoD model assumptions and limitations" section in the supplementary material). The ability to generalize across species is particularly useful for species for which data are limited and models are unavailable, as management and policy decisions need to be made even in the absence of complete information.

L85, 334, & 426: Maybe these sections could just be titled "Life-history traits", "Disturbance source characteristics", etc.? I don't think the "The importance of..." part is necessary.

We removed "The importance of..." from the beginning of sections 2, 3, and 4.

L116: It would be nice to have some provided examples of these lasting effects.

We included "(e.g., Deepwater Horizon oil spill in the Gulf of Mexico [35])" at the end of the sentence.

L117-118 & 126: Maybe it would be worth combining these two statements about some individuals in migrating populations not migrating?

We removed "although some individuals in a population may not migrate (e.g., West Indian manatee (*Trichechus manatus*) [44])". The sentence now reads: "Other migratory populations do not have separate foraging grounds and reproductive areas and instead migrate in response to seasonal ecological conditions, such as advancing sea ice and migration of prey (e.g., beluga whales (*Delphinapterus leucas*) [37])."

L142-143: Could add a “single” disturbed area to highlight

We reworded the previous sentence which required this sentence to be worded differently. The sentence now reads: “When incorporated into a PCoD model, Dunlop *et al.* [41] found that similar behavioural responses to a simulated 10-day seismic survey during peak migration had negligible effects on female body condition and population growth.”

L168-177: Many PCoD models do not explicitly consider animal movement in their simulations. It would be nice to briefly describe to what extent (and under which scales) has movement actually been considered in PCoD models and what scales of movement are important for measuring disturbance responses.

We included the following sentences at the end of the paragraph to address the reviewer’s comments: “PCoD models can be spatially explicit, using both coarse- [13] and fine-scale [49] movement data, or spatially implicit, with movement data reflected in activity-budgets [12] or not included at all [15]. Ultimately, the scale of movement necessary to assess risk depends on the target population and the management or policy issue being addressed.”

L187: Could be good to specify such as “These sensitivities to disturbance in income breeders...”

We added “in income breeders” to the sentence. The sentence now reads: “As such, these sensitivities to disturbance in income breeders can lead to declines in offspring recruitment and overall population size [12].”

L188-189: Is this statement supposed to be related to the lactation period? As all marine mammals can store lipids for later use to some degree, is this intended to state that capital breeders may be less sensitive to foraging losses during the lactation period as lipid used for this process has already been stored? Would be good to clarify.

We added “particularly during the lactation period” and “rely on energy that has already been stored” to the sentence. The sentence now reads: “In contrast, capital breeders are less sensitive to short-term foraging losses, particularly during the lactation period, because they rely on energy that has already been stored [13, 52].”

L200: Could add that this could ultimately impact population abundance.

We added “and, ultimately, population abundance” to the sentence. The sentence now reads: “These models show that reduced foraging opportunities can delay sexual maturity or age at first reproduction [15, 54, 55] and increase the interval between reproductive events [15, 54], which could impact a female’s lifetime reproductive output and, ultimately, population abundance.”

L204-207: But it should be highlighted that we know very little about the specifics of how and when these decisions are made in marine mammals, particularly cetaceans.

At the end of the paragraph, we added “However, how and when these thresholds are reached is poorly understood.”

L231-232: When known? These predictions may be very tough to get for many species/environments.

We added “when known” to the sentence. The sentence now reads: “Additionally, when preparing a risk assessment for a long-term activity, environmental conditions that affect prey availability and marine mammal distribution should be considered when known (see *Environmental conditions*).”

L236: Instead of basal metabolism I would say survival as it seems that this here includes costs additional to true basal metabolism, including activity, thermoregulation, feeding, etc.

We removed “basal metabolism” and added “survival” to the sentence. The sentence now reads: “Body size profoundly influences marine mammal life-history strategies because it affects the rate at which energy is acquired from the environment and how it is allocated to growth, reproduction, and survival [62].”

L243: they require a “relatively” higher resource acquisition rate, but not necessarily absolutely unless this rate is per unit time and mass.

We added “relatively” to the sentence. The sentence now reads: “Smaller individuals or species expend more energy per unit mass than larger ones and thus require a relatively higher resource acquisition rate to meet their metabolic demands.”

L256-258: Though small animals need relatively more food per unit mass they also require less food in total, for periods when food is limited smaller animals may more easily be capable of meeting their energetic needs. There are many benefits to being small which should also be discussed in this section. See: Goldbogen, J.A., Cade, D.E., Wisniewska, D.M., Potvin, J., Segre, P.S., Savoca, M.S., Hazen, E.L., Czapanskiy, M.F., Kahane-Rapport, S.R., DeRuiter, S.L. and Gero, S., 2019. Why whales are big but not bigger: Physiological drivers and ecological limits in the age of ocean giants. *Science*, 366(6471), pp.1367-1372.

We added a sentence to the second paragraph in the “Body size” section to capture the important point made by the reviewer. The sentence reads: “However, when prey is reduced or limited, smaller-bodied species may be better able to meet their energetic needs than larger ones because they require less food in total [64].” Other benefits to being small, particularly at the population level (using harbour porpoises as an example), are captured in the “Pace of life” section.

L271-273: They also have additional costs of growth.

We added the following sentences at the end of the paragraph to capture the reviewer's comment: "Reduced energy acquisition during this important developmental period can also affect the amount of energy allocated to growth. While individuals may be able to compensate for slowed growth over time, their lifetime reproductive output could be impacted [78]."

L276: Maybe "may be exposed" rather than just "present"?

We removed "present" and added "exposed" to the sentence. The sentence now reads: "Understanding which species and life stages may be exposed can help assess which populations may be most sensitive to a disturbance-inducing activity."

L311: Would be nice to reference Read & Hohn 1995 here.

We added Read and Hohn (1995) as a reference at the end of this sentence.

L315: Citation needed?

We added the following reference to the end of the sentence: Rojano-Doñate, L, McDonald, BI, Wisniewska, DM, Johnson, M, Teilmann, J, Wahlberg, M, Højer-Kristensen, J & Madsen, PT. 2018 High field metabolic rates of wild harbour porpoises. *J. Exp. Biol.* 221, 1-12.

L315-317: Can cite Nabe-Nielsen et al. 2018 here as the bounce back after disturbance is visible in their porpoise simulations.

We added Nabe-Nielsen et al. (2018) as a reference at the end of this sentence.

L325-330: This feels a bit tagged on the end, also if this is tied to body condition estimates, I would assume that this sort of pattern would still come out of many of the cited models, but not if it is related to differences in behavior, genetics, etc.

We moved these sentences to the "PCoD assumptions and limitations" section in the supplementary material where we discuss how most PCoD models to date have yet to consider individual heterogeneity.

L329-330: What would be the population level implications of this finding?

We modified one of the sentences to describe the population-level implication of this finding. The sentence now reads: "Thus, some females may be particularly robust and, as a result, a disturbance impacting these females may have a more limited effect on the population as a whole."

L337: Nature is a bit vague, maybe it would be good to provide an example or short description here.

We added some examples of what we mean by “nature of the disturbance source” to the sentence. The sentence now reads: “For example, the spatial and temporal features and nature of the disturbance source (e.g., type (sonar)), operational characteristics (intensity, frequency), and behaviour (moving, stationary)) can interact with life-history traits and other contextual factors to influence the probability and severity of individual responses [94].”

L345-348: This sentence could be reworded to be more concise.

We reworded this sentence to be more concise. The sentence now reads: “For example, simulations carried out by Pirootta *et al.* [61] found that disturbance within important foraging areas had a more dramatic effect on adult female northern elephant seal energy budgets than a similar disturbance located in less important habitat within the population’s range.”

L353: Also important that the similar habitat is of a sufficient area.

We added “of sufficient area” to the sentence. The sentence now reads: “Ultimately, the magnitude of any effect will depend on whether similar habitat of sufficient area is available within the population’s range, as well as the temporal characteristics (see *Duration and frequency*) and nature of the disturbance source and the exposure context (see *Nature and context*).”

L365: "consider"

We replaced “considers” with “consider” in this sentence. The sentence now reads: “For activities that span decades (e.g., offshore wind farms), these static spatial management tools may be less effective unless they consider ecological shifts in response to environmental variability and climate change (e.g., IMMA designations include 10-year review periods to account for climate change-related shifts [97]).”

L368-381: How can these technologies be paired with PCoD models?

Please see response to comment, above, about L168-177.

L384-390: These two sentences have quite a few subordinate clauses in the middle of the sentence. Some rewording could make for smoother reading.

We agree with the reviewer’s comment. To increase readability, we removed the subordinate clauses and broke up the last sentence into two sentences. These sentences now read: “For example, simulations conducted by New *et al.* [7] predicted that an increase in the number of disturbance days would lead to a decline in southern elephant seals’ lipid mass and, subsequently, a decrease in pup weaning mass and survival. They

also found that the predicted decrease in pup survival resulting from a prolonged disturbance (i.e., reducing the duration of a female’s foraging trip by half) in any one year had seemingly minor impacts on the population. However, the effects of repeated exposures over a 30-year period led to a substantial decline in population size.”

L404-405: Second “inform” isn’t needed.

We removed the second “inform” from this sentence. The sentence now reads: “Such information can support activity planning and area-specific caps on disturbance-inducing activities, especially within biologically important habitats [11, 57].”

L407: How are disturbances modelled to impact individuals in these different models (e.g., halting energy intake)? What are the common approaches?

In the “Introduction,” we describe how most implementations of the PCoD framework have focused on bioenergetics and “...changes in a female’s time-energy budget concerning lost foraging time, the subsequent effects on energy delivery from mother to offspring, and the cascading long-term impacts on the population [11-13].”

L416-420: This phrasing implies that received level doesn’t factor at all into responses.

We removed “other than the received sound level” and included the word “additional” to the sentence to capture that received level (as described in the previous paragraph) does factor into responses. The sentence now reads: “An individual’s propensity to respond and the severity of the response likely depend on additional, intrinsic factors.”

L441: “sensitivity to disturbance”.

We added “to disturbance” to this sentence. The sentence now reads: “As a result, strategic planning for the timing of disturbance-inducing activities relies upon understanding the links between abiotic and biotic factors that drive marine mammal sensitivity to disturbance.”

L451-454: What about seasonal or temporal variations in energy intake needs which are not related to reproduction but instead the environment? e.g. the seasonality in energy balance presented in: Gallagher, C.A., Grimm, V., Kyhn, L.A., Kinze, C.C. and Nabe-Nielsen, J., 2021. Movement and seasonal energetics mediate vulnerability to disturbance in marine mammal populations. *The American Naturalist*, 197(3), pp.296-311.

We incorporated text in this paragraph to capture the temporal/seasonal variations in energy intake that could affect an individual’s energy balance. For capital breeders, the sentence reads: “Because there is a limited period to acquire energy, PCoD models show that disturbance-inducing activities that reduce foraging time can affect an individual’s energy balance and thus reproduction and survival [11, 13]. However, the magnitude of the effect will likely depend on the proximity of the disturbance source to important foraging areas, as shown in some PCoD models [13, 61], and whether the disturbance

coincides with periods of increased energy intake [118].” For income breeders the sentence reads: “However, PCoD models show that some populations may be more spatially and/or temporally restricted in their ability to adapt to disturbance-induced changes in foraging during periods of low prey availability [15] and increased energy intake [32].” We included Gallagher et al. (2021) as the reference used for income breeders.

L479-481: Just influences? Would be nice to be more specific here.

We replaced “influences” with “mediated”. The sentence now reads: “For example, Pirotta *et al.* [13] found that, during an ENSO event, the location, duration, and frequency of a simulated disturbance mediated the cumulative effect on blue whale vital rates.”

L514: Were these combined effects synergistic?

Combined effects that are larger than in isolation is what would be normally defined as synergism. The interpretation of terms like synergism (and antagonism) are conflicting across disciplines and depend on how “additivity” is defined, as discussed in the National Academy of Sciences (NAS, 2017) report on the population consequences of multiple stressors (PCoMS). The NAS (2017) report provides a review of meta-analyses that investigated the cumulative effects of multiple stressors and concluded that there are few situations where one can confidently assume that the effects of multiple stressors are additive, and this could lead to an underestimation or overestimation of their cumulative impact. As such, PCoD/PCoMS research is shifting away from using such terms, which is why “combined effects” is used in the sentence. An argument for moving away from terms like synergism and antagonism will be made in an upcoming paper on the PCoMS framework.

L530: This section could be a bit expanded a bit to be more specific about identified data gaps across the different modelling exercises. It would be interesting to see for models in which sensitivity analyses were carried out, if there were any common themes identified for types of data that these sorts of models are particularly sensitive to.

We appreciate the reviewer’s comment. Efforts are currently underway to systematically capture some of the major gaps within the PCoD/PCoMS framework. For example, there is a team working on identifying the data gaps in the bioenergetics pathway to better inform PCoD models. In addition, a new PCoMS project is currently underway that is also identifying data gaps. As such, we included the following sentence in the “Data gaps and future priorities” section: “Efforts are currently underway to systematically evaluate these gaps and inform future research to better parameterize PCoD models.”

Regarding sensitivity analyses, not many PCoD models have explored sensitivity explicitly. Anything related to the prey/environment seems to have very strong effects on model outcomes, as well as to the exposure rates, energetics, amount of time spent feeding (i.e., activity budgets), individual morphology, and context-dependency of behavioural responses (e.g., see Pirotta et al. (2018), McHuron et al. (2018), Gallagher et

al. (2021)). In the “PCoD model assumptions and limitations” section in the supplementary material, we discuss how sensitivity analyses can be used *post hoc* to quantify the uncertainty surrounding model inputs and describe some of the common parameters that models are sensitive to.

L541-542: I would say that for the vast majority of populations this information isn't known, what are the implications of that? and how do we get around this issue?

We added the following to the paragraph to address the reviewer’s comment: “In the absence of demographic information for the target population or a related species, first principles can be used to predict how the population may respond to disturbance [135]. For example, Natrass and Lusseau [135] demonstrate how a basic understanding of a species’ physiology and the productivity dynamics of the environment can be used to estimate a population’s resilience to disturbance. Many of the general principles explored in this synthesis can help inform such an assessment.”

Figure 1: It is a little difficult to read the colored text (particularly the green). I would use bold to make this more legible for those of us with inferior vision.

We appreciate the reviewer’s feedback. To improve readability, we decreased the brightness, saturation, and hue of the green colour and bolded the coloured text.

Figure 2a: I find it a bit challenging to read this figure. I think that, as is, it doesn’t communicate the main points effectively. In the Nomadic and Migratory populations it is unclear whether tracks are from a single or multiple individuals. In the original plot in Costa et al. 2016 the individual tracks were color coded so it was clear, though in the current plot it is already difficult to tell the difference between the pink and orange color at the current size. Is it necessary to have two disturbance zones? It could be nice to have a single zone shown, maybe as a fill with high transparency, and have the individual tracks colored as in the original figure to make clear the different number of individuals being disturbed.

We appreciate the reviewer’s feedback. We removed the smaller/second disturbance zone, increased the transparency of the remaining disturbance zone (renamed “Proposed Activity” to mirror Figure 3), and varied the colour of the individual tracks to make it clear that different individuals may be disturbed.

Figure 3: It would be nice to include variability in this visual as well around the start and end of each period to encourage the consideration of peak periods if relevant - as in the Cornell birds of the world annual cycles. Also I'm not sure that having maintenance here helps anything since it is visualized as constant year round.

We appreciate the reviewer’s feedback. We incorporated variability around the start and end times for each life stage and removed maintenance from the reproductive cycle plot. To show variability, we used a solid colour to indicate when the majority (or peak number) of individuals would be in a specific stage and a striped pattern to show variability in the start/end of each life stage.

Appendix B

UNIVERSITY OF CALIFORNIA, SANTA CRUZ

BERKELEY • DAVIS • IRVINE • LOS ANGELES • MERCED • RIVERSIDE • SAN DIEGO • SAN FRANCISCO

SANTA BARBARA • SANTA CRUZ

DEPARTMENT OF ECOLOGY & EVOLUTIONARY BIOLOGY
DIVISION OF PHYSICAL & BIOLOGICAL SCIENCES
COASTAL BIOLOGY BUILDING
130 MCALLISTER WAY
SANTA CRUZ, CALIFORNIA 95060

July 28, 2021

Dear Editorial Board,

We are pleased to submit our revised manuscript “**Emerging themes in Population Consequences of Disturbance models**” (RSPB-2021-0325.R1) for publication as an evidence synthesis in *Proceedings of the Royal Society B*.

We are grateful for the final, thoughtful reviews provided by the referees. We have incorporated the requested revisions and responded to each comment in the postscript of this letter, in blue.

I can be reached by email (kelly.a.keen@gmail.com) or by phone (+1 215 287 8530). My co-authors' email addresses are listed below:

Roxanne Beltran: roxanne@ucsc.edu

Enrico Pirotta: pirotta.enrico@gmail.com

Daniel Costa: costa@ucsc.edu

Yours sincerely,

Kelly Ann Keen

PhD Student

Ecology and Evolutionary Biology

University of California, Santa Cruz

Reviewer(s)' Comments to Author:

Referee: 1

Comments to the Author(s)

I can see the authors' great effort to address my concerns. I do not have any further concerns, and would like to see this article published as it is.

We appreciate the reviewer's thoughtful comments and kind words about the manuscript.

Referee: 3

Comments to the Author(s)

Dear authors,

After reading through the response to reviewers and edited version of the manuscript, I find this revised version to be much improved, and greatly appreciate the author's responses to the various concerns. The text is much tighter and writing much more focused on the synthesis of PCoD results in this version, which makes for smoother reading. I especially appreciated the additional sections related to Applicability to other species and PCoD model assumptions and limitations.

We appreciate the reviewer's thoughtful comments and kind words about the manuscript.

I have only a few remaining comments:

L178 - Reproductive strategy section: I am missing in this section some mention of offspring provisioning. The mother's susceptibility to disturbance due to high lactation costs may differ if offspring have high levels of provisioning as they reach weaning age.

We capture the reviewer's comment on lines 212-220 of the manuscript: "Model simulations have also demonstrated that the timing of a disturbance-inducing activity during these sensitive states influences whether a female can compensate for reduced or lost foraging. For income-breeding California sea lions (*Zalophus californianus*), simulations carried out by McHuron *et al.* [60] found that the costs associated with nursing a pup were much greater during late lactation than early lactation because the total energy delivered to the pup increased as the pup grew. In contrast, for capital-breeding northern elephant seals (*Mirounga angustirostris*), Pirotta *et al.* [61] found that simulated females could better compensate for disturbance during the first phase of their 8-month foraging trip if the disturbance was not severe."

L275: The "As a result" implies that PCoD models are considering factors like oxygen-carrying capabilities and the impacts of experience on an animal's ability to cope with disturbance, and that emerging high rates of starvation for juveniles and young females in PCoD models are due to these factors, which, to my knowledge, isn't the case.

We removed "As a result" and added "Due to their small body size".

L411 - Nature and context of disturbance section: Since dose-response curves are mentioned in the conclusion, it would be nice if they were mentioned here in this relevant section as well.

We added the following sentence in the first paragraph (beginning on line 419): “This research has led to the development of analytical tools such as dose-response functions, which provide a framework for relating an individual’s probability of responding to some metric of exposure (e.g., received sound level) [104].” We also added “(e.g., via improved dose-response functions)” to line 439. The sentence now reads: As outputs of behavioural response studies become available, they should be incorporated into PCoD models and risk assessments (e.g., via improved dose-response functions) and used to develop mitigation and monitoring protocols that validate predictions [96, 104].

In the Appendix PCoD model assumptions and limitations section: It would also be nice to see some mention of how most PCoD models generally only focus on single species and do not include important interspecific interactions or competition. Additionally interactions and competition within species are generally not modelled but can have important implications for the prediction of disturbance effects (see Hin, V., Harwood, J. and de Roos, A.M., 2021. Density dependence can obscure nonlethal effects of disturbance on life history of medium-sized cetaceans. *PLoS one*, 16(6), p.e0252677.)

In the first paragraph, we added “density dependence” to the list of factors that affect PCoD model outcomes and added Hin et al 2021 as a reference at the end of the sentence.

We also edited the third paragraph to capture interspecific interactions/competition. The paragraph now reads:

Most PCoD models have also only considered sources of disturbance in isolation. However, multiple sources of anthropogenic and environmental disturbance are likely to act concurrently in any one area. Similarly, PCoD models currently focus on a single species and do not include critical interspecific interactions, such as competition, that may affect the response to disturbance. Consequently, this requires a non-trivial extension of the PCoD framework to capture the effects of multiple ecological interactions and disturbance sources or stressors [168]. A conceptual framework for the Population Consequences of Multiple Stressors has recently been considered [168], but research is just beginning. Future research will provide valuable information regarding the functional links between multiple ecological interactions and sources of disturbance and their potential population-level effects (see *Data gaps and future research priorities*).

Overall, I believe that with these minor changes this timely review will be suitable for publication in Proc B.

Referee: 2

Comments to the Author(s)

Excellent revisions and additions. The convergence of comments among the reviewers was notable to me and I think all the additional details in paper, new appendices, and references are really helpful and justified. Thank you for putting so much time and attention into these comments. The paper is much improved, I absolutely support publication, and I think it will be very important in increasing understanding and acceptance of these processes in decision-making. I have one very minor additional suggestion, which is that I think you should do just a bit more both in the introduction and in the discussion to highlight the details and implications in Appendix C and (especially) D. Because this has been a criticism of the PCOD approach (transparency and understanding of limitations/assumptions), I think it should be a bit more explicitly discussed and emphasized in terms of how this paper now provides that clarity and guidance for managers. I also think it should be made clear(er) that because of all the species and contextual differences that are well discussed, this application of the process in risk assessments is not nor ever will be a simple, uniform 'formula' but that it will need to be adapted and tuned in terms of how it is applied. You do say this but just at the end of the intro and again in the discussion, I suggest you provide these messages a little more bluntly and clearly almost speaking directly to the managers. I see these suggestions as literally a few sentences in each section and really think the rest of the paper looks fantastic and ready to go.

We appreciate the reviewer's thoughtful comments and kind words about the manuscript.

Due to space/page limits, we were unable to add a few sentences to each section of the manuscript; however, we did reword/add a couple of sentences in the "Introduction" and "Data gaps and future priorities" sections to capture the reviewer's comments.

In the "Introduction", we added "can influence model predictions but" to line 87. The sentence now reads: "...as well as underlying model assumptions and limitations that can influence model predictions but may not be self-evident to non-specialists or non-statisticians (see *PCoD model assumptions and limitations*)."

In the "Data gaps and future priorities" section, we reworded the second paragraph to read: "The amount of data and processing time required for PCoD models can limit their direct application in decisions about proposed activities. Additionally, due to species and contextual differences, there is not a simple, "one-size-fits-all" approach for applying the PCoD framework in risk assessments. As such, its application will need to be adapted based on the data available and issue being addressed. Collaborations between modellers and wildlife managers can help identify ways to increase model accessibility and adaptability and the outputs necessary to make decisions. Furthermore, model assumptions and their influence on outputs should be made transparent to and considered by end users, such as wildlife managers and policymakers, in subsequent decisions (see supplementary material, *PCoD model assumptions and limitations*).